# From Promise to Practice:
# A Study of Common Pitfalls Behind the Generalization Gap in Machine Learning

**Saeideh Ghanbari Azar**                                    *saeideh.ghanbari@umu.se*
*Department of Computing Science*
*Umeå University*

**Lorenzo Tronchin**                                         *lorenzo.tronchin@umu.se*
*Department of Computing Science*
*Umeå University*

**Attila Simko**                                             *attila.simko@umu.se*
*Department of Diagnostics and Intervention, Biomedical Engineering and Radiation Physics*
*Umeå University*

**Tufve Nyholm**                                             *tufve.nyholm@umu.se*
*Department of Diagnostics and Intervention, Biomedical Engineering and Radiation Physics*
*Umeå University*

**Tommy Löfstedt**                                           *tommy.lofstedt@umu.se*
*Department of Computing Science*
*Umeå University*

**Reviewed on OpenReview:** *https://openreview.net/forum?id=DqWvxSQ1TK*

## Abstract

The world of Machine Learning (ML) offers great promise, but often there is a noticeable gap between claims made in research papers and the model's practical performance in real-life applications. This gap can often be attributed to systematic errors and pitfalls that occur during the development phase of ML models. This study aims to systematically identify these errors. For this, we break down the ML process into four main stages: data handling, model design, model evaluation, and reporting. Across these stages, we have identified fourteen common pitfalls based on a comprehensive review of around 60 papers discussing either broad challenges or specific pitfalls within ML pipeline. Moreover, Using the Brain Tumor Segmentation (BraTS) dataset, we perform three experiments to illustrate the impacts of these pitfalls, providing examples of how they can skew results and affect outcomes. In addition, we also perform a review to study the frequency of unclear reporting regarding these pitfalls in ML research. The goal of this review was to assess whether authors have adequately addressed these pitfalls in their reports. For this, we review 126 randomly chosen papers on image segmentation from the ICCV (2013–2021) and MICCAI (2013–2022) conferences from the last ten years. The results from this review show a notable oversight of these issues, with many of the papers lacking clarity on how the pitfalls are handled. This highlights an important gap in current reporting practices within the ML community. The code for the experiments is available at `https://github.com/SG-Azar/BraTS-ML-Pitfalls-Experiments`.

# 1 Introduction

Machine Learning (ML) is reshaping our world by its unparalleled ability to extract patterns from vast amounts of data. From enhancing the capabilities of self-driving vehicles to improving diagnostic accuracy in medical imaging, the influence of ML is evident in numerous fields and applications. Despite its potentials, there are various tales of ML models failing catastrophically in real-world settings. For instance, after the launch of the IBM's Watson For Oncology (WFO) project in 2013, there were news about its unsafe and erroneous cancer treatment recommendations (Ross & Swetlitz, 2018). Another example is the "Tay" chatbot released by Microsoft that quickly started to produce offensive tweets, highlighting the risks of unsupervised learning from the vast, unfiltered content of the internet (Hunt, 2016).

Recently concerns are raised about potential racial and gender biases in ML-based technology (Buolamwini & Gebru, 2018). These issues have become even more complicated by the recent advancements of generative, and large language models. For instance, Stable Diffusion is a model that generates images based on written prompts. A study analyzing over 5000 images generated by Stable Diffusion, revealed significant racial and gender disparities in the outputs, with the apparent tendency in the model to associate high-paying jobs like CEOs, doctors, and lawyers to lighter-skinned men (Nicoletti & Bass, 2023). A further level of complexity can be added if these outputs are used to create datasets for other ML models, creating models with various types of unknown biases.

Moreover, the reliability of developed ML model in critical applications remains an issue. For example, a review by Roberts et al. (2021) analyzed 62 quality screened studies, selected from an initial pool of 2212 papers, that all aimed to develop ML models for Covid-19 detection from CT scans. They concluded that all studies had methodological errors or biases and were therefore not reliable enough to be used clinically. Another study by Yu et al. (2020) compared the performances of the winning models from the Kaggle Data Science Bowl (AJ_Buckeye et al., 2017) competition. Their analysis revealed a notable discrepancy in performance when these models were applied to different test sets, raising questions about their generalizability.

There thus seems to be a large gap between claimed research performance of ML models and their real-world performance. But why does this discrepancy exist? To some extent, the limitations in generalizing from one situation to another are inherent in empirical research, but often these failures can be attributed to systematic errors and issues in the ML process.

Many studies have been conducted in recent years that investigate these issues and try to pinpoint recurring pitfalls in different stages of the ML process. Since this process is very elaborate, with many complicated and interdependent steps, most such studies focus their analysis on a single stage or a specific pitfall. For instance, Paullada et al. (2021) surveyed issues in data collection and usage in ML. Specifically, their main concern and focus was on biased datasets, annotation problems, spurious correlations within datasets, and misuse of data that contains sensitive information. There are several studies that have addressed the generalization problem in ML models and ascribe it to spurious correlations (Wallis & Buvat, 2022; Geirhos et al., 2020; Lapuschkin et al., 2019). For example, Geirhos et al. (2020) defined shortcut learning as learning decision rules that are not relevant to the prediction task. For instance, the model may use some correlations from the background to classify objects, instead of using the characteristics of the object itself. A notable example of this phenomenon was presented by Ribeiro et al. (2016) where a ML model distinguished between wolves and huskies based on the presence of snow in the background rather than the animals' physical characteristics. Geirhos et al. (2020) argue that models that have learned to shortcut will perform well under laboratory conditions but cannot generalize to real-world problems.

The evaluation of a developed model lie at the heart of the ML process but is unfortunately prone to many pitfalls, and has therefore been the focus of many studies. For instance, Liao et al. (2021) referred to evaluation as the "stumbling block" in many ML subfields. Based on a meta-review that they performed on 107 papers, the common evaluation failures were divided into two categories: internal validity, concerning evaluation with a single benchmark, and external validity, involving assessments in broader scenarios outside the benchmark. Raschka (2018) presented an extensive review on common evaluation and model selection techniques and gave recommendations for the best evaluation practices in ML research and applications.

Among many other subjects, he comprehensively covered different variations of cross-validation and statistical tests. Other studies have focused on metrics-related issues and pitfalls, and have given recommendations for the correct use of metrics for model evaluation (Blagec et al., 2020; Reinke et al., 2023).

Although focused studies provide much insight into the details of parts of the ML process, there are also studies that have addressed the whole ML pipeline. However, such studies have mostly been performed within specific fields of study. For instance, Varoquaux & Cheplygina (2022) reviewed some of the roadblocks for developing and assessing ML models in medical imaging. They covered data limitations and biases and discussed improper evaluation procedures such as information leakage and incorrect use of metrics. They also mentioned publication and reporting malpractices. Lones (2021) covered the entire ML pipeline and presented a simple and comprehensive set of recommendations for correct and incorrect practices in ML. Although the paper was written from a scholarly perspective and experience, it is not a technical explanation but rather a narrative list of *Dos and Do Nots* for ML beginners. Hullman et al. (2022) compared the pitfalls in learning from data in psychology and ML. While drawing some insightful ideas from comparing the two fields, they also summarized the common errors that occur in different stages of a ML pipeline. There are also studies that focus on best practices and common pitfalls when using ML in genomics data science (Teschendorff, 2019; Whalen et al., 2022), chemistry (Artrith et al., 2021), business (Van Giffen et al., 2022), and mental health research (Tornero-Costa et al., 2023).

Despite the existing literature and the sometimes simple nature of some of the pitfalls, these errors are still abundant in the ML research publications. For instance, although using an independent test set for evaluation may seem trivial, review studies verify that this is often overlooked in many studies (Poldrack et al., 2020; Tornero-Costa et al., 2023; Tampu et al., 2022). This prevalence of common errors indicates a broader issue: insights about these pitfalls are scattered across the literature, making it difficult for researchers to get a full understanding of the pitfalls from bits and pieces spread across different studies. Hence, a complete, easy-to-use list may help researchers and practitioners see all the potential mistakes in one place.

Recognizing this gap, we have extensively reviewed numerous papers that discuss pitfalls, errors, and issues contributing to the generalization gap. We have curated and categorized a comprehensive list that encapsulates the most pivotal errors and challenges faced in the four key stages of the ML pipeline. We emphasize that the scope of this paper is restricted to generalization as a goal in ML, focusing on technical pitfalls that directly impact a model's ability to generalize. While acknowledging that ML includes a variety of other goals like explainability, fairness, and robustness, we do not cover these aspects in this paper. Additionally, these pitfalls are mostly discussed from a technical perspective rather than from organizational and cultural perspectives.

We also show the impact of these pitfalls through examples, using the Brain Tumor Segmentation (BraTS) dataset (Baid et al., 2021; Menze et al., 2014), that illustrate what can go wrong and how these mistakes can affect the final outcomes. Moreover, to study how clearly recent ML research addresses these pitfalls, we reviewed 126 papers from the International Conference on Computer Vision (ICCV) and the International Conference on Medical Image Computing and Computer Assisted Intervention (MICCAI) conferences, spanning the last ten years. Our goal was to see how often these papers clearly acknowledge or overlook these pitfalls, providing insight into the lack of clarity in the reporting. With this paper, we hope to contribute to the overall improvement of ML results and help to establish more rigorous and error-free practices. In short, we make the following contributions:

- Review numerous papers to identify and categorize fourteen common pitfalls that affect the generalizability of ML models.

- Demonstrate through experiments how these pitfalls can skew results and impact model performance.

- Analyzes 126 papers from ICCV and MICCAI conferences to assess how recent ML research addresses these pitfalls, highlighting a prevalent lack of clarity in reporting.

The remainder of the paper is structured as follows: Section 2 details the common pitfalls associated with four main steps of the ML pipeline. Section 3 describes three experiments conducted to illustrate the impact of selected pitfalls on ML model performance. Section 4 reviews papers from the MICCAI and ICCV

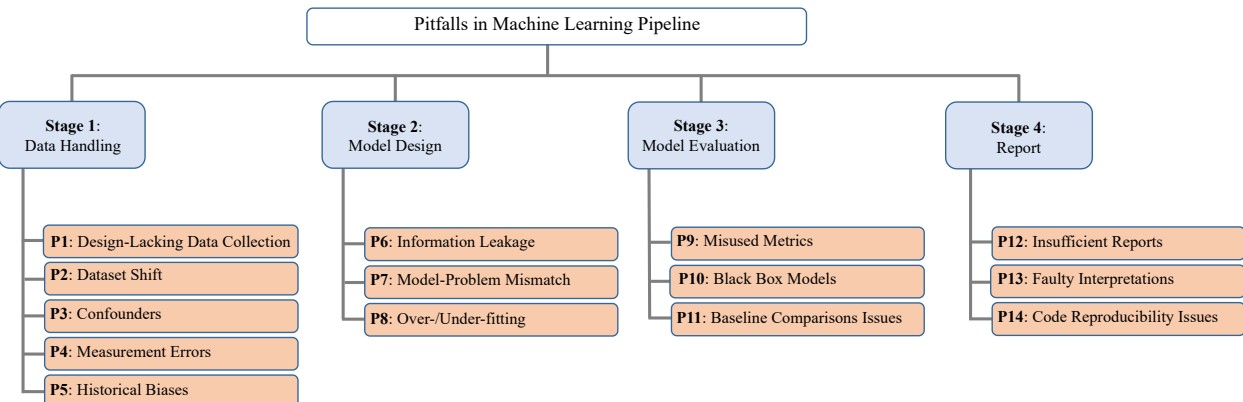

Figure 1: Taxonomy of common machine learning pitfalls categorized by the four stages of the machine learning pipeline.

conferences, analyzing how these pitfalls are addressed in current research. Finally, Section 5 concludes the paper.

## 2   The ML Pipeline

Given a collection of input-output data pairs, drawn from some distribution, the goal of supervised ML is to create a function that predicts the outputs based on the input values. Collecting and processing these data is the first step in any ML project. The second step is to design and train a model for the problem at hand. In this step, by feeding data into the model and adjusting its parameters to fit the data, the function gets better at making predictions for new data points. This iterative process makes use of a loss function, which quantifies how far off the model's predictions are from the actual outcomes. Since this process is highly data-centric, there's an inherent risk of overfitting to the observed data, which leads to failures to generalize to new data. To estimate the generalization gap and to avoid overfitting, a very common approach is to divide the sampled data into three parts with training, validation, and test data. The model parameters are updated based on the training data set. Model selection is performed and overfitting is monitored using the validation data set. The test set is used in the third step of a ML process to estimate the generalization performance of the model. Finally, the results and conclusions are communicated in the fourth and final step of a ML project. However, across these steps there are many potential pitfalls, and systematic errors may arise that lead to a disparity between observed performance during development and the model's actual performance in the real-world, *i.e.*, the generalization gap.

In the following, we present fourteen specific pitfalls that can lead to errors in the overall ML process and address each in turn in the following subsections, where we have designated them as P1 through P14 namely, P1: Design-Lacking Data Collection, P2: Dataset Shift, P3: Confounders, P4: Measurement Errors, P5: Historical Biases, P6: Information Leakage, P7: Model-Problem Mismatch, P8: Over-/Under-fitting, P9: Misused Metrics, P10: Black Box Models, P11: Baseline Comparisons Issues, P12: Insufficient Reports, P13: Faulty Interpretations of the Results, and P14: Code Reproducibility Issues. This list was extracted based on a comprehensive review of around 60 papers discussing either broad challenges or specific pitfalls within ML pipeline. This review began with targeted searches on "best practices in machine learning" and "machine learning pitfalls", and was supplemented by examining references within these papers as well as the experiences of the authors. Table 1 lists the 60 papers reviewed in this study, along with the specific pitfalls each addresses. Note that while some papers provide detailed discussions of a pitfall, others may only briefly mention it. Figure 1 shows a taxonomy of these pitfalls categorized by the four stages of the ML pipeline.

Table 1: A list of the reviewed papers contributing to the identified pitfalls.

| Reference | Pitfalls |
|---|---|
| Quiring et al. (2022) | P2, P3, P4, P6, P9, P11, and P13 |
| Kelly et al. (2019) | P2, P3, P5, P9, P10, and P11 |
| Domingos (2012) | P6, P7, P8, and P13 |
| Biderman & Scheirer (2020) | P1, P2, P4, P5, and P13 |
| Suresh & Guttag (2021) | P2, P4, P5, P6, and P9 |
| Poldrack et al. (2020) | P2, P6, P8, P9, and P13 |
| Tornero-Costa et al. (2023) | P2, P5, P10, P13 |
| Varoquaux & Cheplygina (2022) | P2, P4, P5, P6, P9, P11, P12, and P13 |
| Teschendorff (2019) | P2, P3, P6, P8, and P13 |
| Lones (2021) | P2, P6, P7, P8, P11, P12, and P13 |
| Paleyes et al. (2022) | P2, P4, P5, P7, P8, and P9 |
| Baier et al. (2019) | P2 and P7 |
| Walsh et al. (2021) | P2, P4, P6, P8, P9, P10, and P12 |
| Bluemke et al. (2020) | P2, P6, P9, P10, P12, P14 |
| Hullman et al. (2022) | P2, P3, P4, P9, P11, P12, P13, and P14 |
| Artrith et al. (2021) | P2, P7, P12, and P14 |
| Wujek et al. (2016) | P2, P4, P8, and P13 |
| Mojica-Hanke et al. (2023) | P2, P4, P7, P8, P12, and P14 |
| Van Giffen et al. (2022) | P2, P5, and P7 |
| Riley (2019) | P3, P6, and P9 |
| Walsh et al. (2016) | P2, P3, P6, P7, P8, P9, and P11 |
| van Royen et al. (2023) | P2, P7, P12, P13, and P14 |
| Whalen et al. (2022) | P2, P3, and P6 |
| Vandewiele et al. (2021) | P6 |
| Reinke et al. (2021) | P9 |
| Reinke et al. (2023) | P9 |
| Maleki et al. (2022) | P6 |
| Gencoglu et al. (2019) | P11, P12, P13, and P14 |
| Kapoor & Narayanan (2022) | P6 |
| Ng et al. (1997) | P8 |
| Moreno-Torres et al. (2012) | P2 |
| Santa Cruz et al. (2021) | P2, P3, and P4 |
| Wolff et al. (2019) | P2 and P13 |
| Lapuschkin et al. (2019) | P3 |
| Simkó et al. (2024) | P14 |
| Belkin et al. (2019) | P8 |
| Mongan et al. (2020) | P12 |
| Thomas & Uminsky (2022) | P9 |
| Raschka (2018) | P7, P8, and P11 |
| Paullada et al. (2021) | P2, P3, P4, and P5 |
| Molnar et al. (2020) | P7, P8, P11, and P13 |
| Varoquaux & Colliot (2023) | P6, P9, and P11 |
| Roberts et al. (2021) | P2, P4, P10, P11, P12, and P14 |
| Tampu et al. (2022) | P6 |
| Wallis & Buvat (2022) | P3 |
| Nsubuga et al. (2024) | P2 |
| Liao et al. (2021) | P4, P6, and P11 |
| Geirhos et al. (2020) | P3 |
| Mikołajczyk-Bareła & Grochowski (2023) | P2, P4, and P5 |

Table 1 continued from previous page

| Reference | Pitfalls |
| --- | --- |
| Mehrabi et al. (2021) | P2, P4, and P5 |
| Sanchez et al. (2022) | P3 and P13 |
| Haibe-Kains et al. (2020) | P12 and P14 |
| Pineau et al. (2021) | P12 and P14 |
| Lehman et al. (2020) | P9 |
| Stevens et al. (2020) | P12 |
| Howard & Borenstein (2018) | P2 and P5 |
| Rudin & Radin (2019) | P10 |
| Raimondi et al. (2021) | P3 |
| Santa Cruz et al. (2021) | P3, P5, P10, P11, and P12 |
| Dockès et al. (2021) | P2 |

## 2.1 Data Handling

Data is the basis for all ML models, and the quality of the model depends strongly on the quality and diversity of the data. A small error in the data handling can not only propagate through the entire ML process, but when such data are shared within the scientific community, they also have the potential to misdirect future work and by that our collective understanding and the conclusions we draw. In the following we elaborate on five pitfalls related to the data.

### P1—Experimental Design–No Systematic Data Collection

Studies in ML are often observational and retrospective, meaning that we as analysts lack control over the data that we work with. It is important to know that observational studies do not provide the same reliability as designed studies, *e.g.*, as randomized control trials, and therefore requires the analyst to be more careful in the analysis and when drawing conclusions from the resulting models.

The idea that one should randomly assign treatments to groups and to design experiments was put forth in the seminal works of Peirce (Peirce & Jastrow, 1885; Peirce, 1876). This was the seeds of what became *experimental design*. The work of Fisher (Fisher, 1992; 1936) was very influential; he developed these ideas further and also popularized experimental design among researchers. One of the main ideas Fisher introduced was to make experiments with systematic changes to the variables controlling the experiment in such a way that the source of a change in the output variable could be reliably detected. He also realized the importance of and directly connected statistical methodology to the design of experiments. The idea of sequential experiments, performing experiments that depend on previous experiments already performed, was pioneered by Wald (1992).

The purpose of experimental design is to plan an experiment such that data is collected and when subsequently analyzed would result in valid, meaningful, and objective conclusions (Montgomery, 2020). The statistical literature on experimental design is vast and span over more than one and a half century, but the ML literature covering this topic is instead rather sparse—at least when it comes to designing experiments and collecting data.

The main methodology from statistical experimental design that has carried over to ML includes active learning (Ren et al., 2021) and sequential experiments (Bouneffouf et al., 2020), but A/B testing would of course also fall in this category of methods. For instance, in active learning, a technique like uncertainty sampling is used where a model focuses on learning from data it is least certain about (Ren et al., 2021). In sequential experiments, bandit models are a key example, where decisions are iteratively improved based on the outcomes of previous choices (Bouneffouf et al., 2020).

Thus, while methods from design of experiments are sometimes used in ML, they are not widely used. In fact, they are rarely used when collecting data, and rarely used to guarantee certain properties of the model or results. Instead, data are often already existing when a ML project starts, or the procedure to collect

data is a matter of getting as much as possible, rather than collecting data points with certain properties. This may lead to more data being collected in the end, but the quality of the data may be very low. In these situations, the data are likely not *Independent and Identically Distributed* (IID) and may also contain many different kinds of biases. Both issues may lead to models and predictions that suffer, and in the end contribute to the generalization gap. For instance, ImageNet (Deng et al., 2009), a very important dataset in the recent ML advancements because of its size and supposed variability, has been shown to contain various biases that affect the generalizability of models trained on it (Torralba & Efros, 2011). Similarly, in the famous MNIST dataset, the first 5000 images in the test set are designated as "easy" and the last 5000 images as "difficult", a distinction not commonly reported and that doesn't seem to be well-known (LeCun et al., 1998; LeCun, 1998).

Although experimental design enables relevant data collection and experimentation, it is not always scalable due to limited resources. On the other hand, observational data can often be obtained and utilized in large volumes, making it particularly valuable for ML models in uncovering patterns across large datasets. However, when relying on observational data, it's crucial to be aware of potential pitfalls, some of which we will discuss in the sections that follow.

In clinical and industrial settings, ML process usually involves gathering and updating datasets in real time, in contrast to academic studies that often rely on pre-existing datasets. This makes data collection more complex. While standardized datasets are useful in academic research, they do not capture the challenges of real-world data collection and modeling. To collect new datasets in practical settings requires more attention to data quality, changes in distribution, and continuous updates in the data, all of which can affect how well models generalize.

While it may not be possible to employ design of experiments in large-scale ML settings (with thousands of model parameters) or in settings with large scale retrospective observational data. However, active learning approaches may still be beneficial both when a large dataset is already available or in sequential experiments when it is possible to select the next set of samples to update the ML model with. This may allow *e.g.*, to either train on fewer samples, to avoid different kind of biases in the training data, or to make sure that the training examples used are representative of general conditions. In both situations, active learning (Li et al., 2024) and adaptive experimental design (Greenhill et al., 2020) may help to reduce the generalization gap, even in situations where we cannot guarantee that the model has certain properties.

**P2—Dataset Shift**

The IID assumption mentioned above is fundamental to ML performance and generalization. It means that each data point is independent and identically distributed, and it is crucial throughout the ML process, from training to testing. This assumption validates the statistical methods we use to learn and evaluate models because it lets us apply statistical tools like the central limit theorem. Essentially, it suggests that a model trained on a dataset should perform similarly on new, unseen data from the same distribution. If this assumption is not met, the model's accuracy and its ability to generalize to new data could be greatly affected.

Once we have training data, we use empirical risk minimization (Shalev-Shwartz & Ben-David, 2014) to find the model parameters that minimize an error with respect to those data. The error estimate over the training data work as a proxy for the performance we hope to achieve over the entire data distribution. If the training data distribution differs from the distribution of the target population, the developed model will not generalize to unseen data from the target population. This phenomenon is referred to as different names in the literature, but here we call it *dataset shift* as in Moreno-Torres et al. (2012). It is important to note that by dataset shift, we do not mean a shift from one completely different dataset to another, but rather a shift from the training or development dataset to the target population. This concept covers issues like:

**Sampling bias:** It happens when the data collected for model training fails to accurately represent the target population (Moreno-Torres et al., 2012). This issue can take various forms, such as over-representation of certain subsets of the data or under-representation of others. A notable example is Amazon's artificial intelligence recruiting tool, which displayed bias against female candidates. This

bias arose because the tool was trained on in-house data consisting of résumés predominantly from men, and did not adequately represent the target population, which for a recruiting tool includes all potential job candidates, regardless of gender (Cooper, 2018). From a fairness perspective, in this example the AI system perpetuated the existing gender imbalances, a point more discussed in P4.

**Imbalanced population:** Sometimes the target population itself consists of minority or majority groups, and even with a perfect sampling, the collected data will have representational issues. For example, in a medical dataset for adults aged 18–40, pregnant women might form only 5% of the population. Despite perfect sampling, the model may still lack robustness for this subgroup due to the limited data available to learn from their specific characteristics (Suresh & Guttag, 2021).

**Non-stationary environments:** If our data come from an environment where the statistical properties are changing in time or space, then a developed model in a certain time or location may no be applicable to new data from another time or location (Moreno-Torres et al., 2012). As an example of non-stationary environments in space, let's assume we have developed a model to segment a body organ based on a dataset from hospital $A$, and now we want to test it on a dataset from hospital $B$, where the standards for the delineation of that organ are different. Similarly, within the same facility, whether a hospital or an industrial setting, equipment upgrades or changes in operational protocols over time can alter data characteristics significantly, leading to a non-stationary environment in time. Another example would be if in the time between when the model was trained on some data and when it is applied to new data, the definition of what should be included in the organ delineation was changed.

Note that in practical settings, it might not be immediately clear what the IID assumption implies in a particular situation. Real-world datasets often suffer from issues such as sampling bias, where the data collected may not accurately represent the target population. For instance, data obtained from different medical centers may vary significantly due to differences in imaging protocols, patient demographics, or other factors, leading to samples that fail with varying degrees to reflect the broader population. Moreover, what constitutes a biased sample may depend on the specific goals of a study. Therefore, it is important to determine the levels across which the model is expected to generalize. Medical imaging data, for instance, contains multiple levels of grouping, from individual slices within a scan to full scans, patients, and hospitals. The generalization requirements vary based on the model's intended application. For instance, if the goal is to generalize to new hospitals, a validation strategy that holds out entire hospitals from training provides a more accurate assessment of real-world performance. Alternatively, if generalization within a single hospital is sufficient, holding out patients may be appropriate. In Experiment 1 in Section 3, we specifically evaluate the model's generalization across hospitals by holding out entire institutions, simulating conditions where the model encounters unseen institutional data.

To handle these problems, several strategies can be used. For example, stratification ensures specific subgroups within a population are adequately represented when splitting the data (Oakden-Rayner et al., 2020). For imbalanced population or labels, techniques like over-/under-sampling can provide a more robust training set (He & Garcia, 2009). Continuous monitoring and model updating are vital in non-stationary environments to ensure the model's relevance over time and across different spaces (Ditzler et al., 2015). Lu et al. (2018) provide an overview of methods for addressing dataset shift, including techniques for both detection and adaptation. For detecting drift, they discuss error rate-based, data distribution-based, and multiple hypothesis test methods. Examples of such methods include Drift Detection Method (DDM) (Gama et al., 2004), Equal Density Estimation (EDE) (Gu et al., 2016), and Hierarchical Change Detection Tests (HCDTs) (Alippi et al., 2016). Once drift is detected, retraining, ensemble methods, and model adjusting are recommended to adjust the model to the new data conditions. Above all, it's important to engage with domain experts; their insights can reveal inherent biases in data collection, performance, and results (Seymoens et al., 2019). Importantly, testing the model on various data and in various environments gives confidence in its performance. In practice, ML benchmarks and competitions, such as *e.g.*, Kaggle, typically provide independent test sets to evaluate models. Although continuous tuning of models on these public test sets may increase the risk of overfitting and leakage, studies by *e.g.*, Recht et al. (2018; 2019); Roelofs et al. (2019) indicate that this risk is often overstated. These studies examined overfitting in popular benchmarks like

CIFAR-10 and ImageNet by introducing newly curated test sets and observed performance drops compared to the original test sets. However, they found that despite the drop in absolute accuracy, the relative ranking of models remained consistent. This suggests that while overfitting to the public test set might occur, it does not drastically affect the relative merits of models. Therefore, these benchmark datasets continue to serve as useful tools for developing models that perform well across different environments.

Note that while the IID assumption and the challenges of dataset shift remain critical considerations in traditional ML models, recent advancements in large, multi-modal, and multi-task models provide new perspectives. Foundation models–large pre-trained models that can be adapted for a wide range of tasks, such as GPT-4, BERT, and CLIP–have achieved unprecedented performance on a broad variety of discriminative and generative tasks (Zhou et al., 2023). Empirical results show that these models perform well under dataset shift, as shown by their state-of-the-art results on several benchmarks, such as the WILDS dataset (Koh et al., 2021; Kumar et al., 2022). These models leverage extensive pre-training and fine-tuning processes to adapt to a wide range of tasks, seemingly relaxing the stringent requirement of the IID assumption by training on enormous amounts of data. Although substantial gaps still remain in their in-distribution and out-of-distribution performance, by definition a step away from the IID requirement, this rapidly growing area of research holds promise for improving model performance and generalization in real-world applications where dataset shift is inevitable.

### P3—Spurious Correlations

ML models typically learn correlations between input and output variables, which can sometimes be influenced by factors unrelated to the target prediction task. According to Ye et al. (2024) spurious correlations in ML are non-causal associations between input features and output labels that a model relies on during training but do not hold consistently across data sets. This can lead to poor generalization even when there is no dataset shift. Spurious correlations can emerge from different sources, including chance occurrences when data is limited or imbalanced. For instance, sampling biases in the development dataset may lead the model to pick up on random patterns that are not representative of the target population. This is closely related to overfitting, which is discussed in the Over-/Under-fitting pitfall (P8). Another source of spurious correlations is confounding variables, or "confounders", that affect both the predictive variables and output variables. When the overall goal is prediction and a high prediction performance, confounders may not be detrimental to this goal, and may sometimes even help. However, if the goal is to explain or understand the underlying relationship between measured variables and the target variable, confounders may introduce biases in both inference and conclusions and can only be properly handled within a causal framework (Pearl & Mackenzie, 2018). As a simple example, suppose we aim to understand the relationship between income and the likelihood of buying a particular product. Education level, in this case, may influence both income and the likelihood of a purchase, making it a confounder that affects the observed relationship.

Two concepts that exemplify cases of unintended reliance on correlations from chance occurrences are the *Clever Hans* decision strategy and shortcut learning. Both involve models leveraging spurious correlations for predictions instead of truly relevant patterns (Geirhos et al., 2020). For instance, visual artifacts can act as proxies for image class detection. A famous such example is the husky-wolf mistake presented by Ribeiro et al. (2016), where snow in the background was a confounding factor that gave the model a shortcut to the image class. Similarly, variations in image resolution can lead models to differentiate classes based on resolution differences rather than the actual content of the images. Lapuschkin et al. (2019) present another example, where a model learned to classify images of horses by detecting a specific tag in the image rather than identifying horse features. They showed that by removing the tag or placing it on other objects, like cars, the model was mislead to incorrectly classify images. Another instance of the Clever Hans effect was explored by Wallis & Buvat (2022), where a bias in a widely-used brain tumor MRI dataset was revealed. Their findings showed that the models could accurately classify tumors, not by analyzing the tumor's characteristics, but rather by exploiting unintended hints introduced in the data preparation stage. Specifically, the position and orientation of the MRI slices in the data were a shortcut to the classification label. In all these situations, the models failed to generalize when tested on data without these spurious correlations.

To mitigate the impact of such correlations, a deep understanding of the problem at hand and the data is crucial. Engaging with domain experts and conducting rigorous data exploration can facilitate the identification of potentially misleading patterns and correlations in the data (Paullada et al., 2021). Once we have this understanding, strategies such as balancing data subsets, standardization, and stratification techniques can help minimize reliance on dataset-specific patterns.

To differentiate between correlation and causation, ML models must be developed within a causal framework. By constructing models under causal assumptions, usually represented as directed acyclic graphs (DAGs) causally connecting variables, causal machine learning allows for the identification of causal relationships and therefore the determination of confounders. This is especially useful when we anticipate a shift in the distribution of data from train to deployment environments. For instance, when we have a non-stationary environment where the patterns and correlations in data may change over time. Traditional models may rely on such correlations, making them struggle with shifts in data. Causal models, however, focus on stable cause-and-effect relationships, allowing them to adapt better to these changes and maintain reliable predictions (Schölkopf et al., 2012).

To address spurious correlations, Ye et al. (2024) outline several key techniques such as data manipulation, representation learning, and learning strategies. Data manipulations techniques, like augmentation, enhance the diversity of the training data to reduce reliance on spurious features. Representation learning approaches, including causal intervention, invariant learning, and contrastive learning, aim to help the model better understand the underlying relationships between variables and become more resilient to spurious correlations. Specifically, contrastive learning helps models distinguish between essential and non-essential features by learning similar representations for same-class samples despite the presence of varying spurious features. Finally, learning strategies are the group of methods that help prevent strong reliance on spurious correlations during the learning process. This may include optimization, ensemble learning, identification then mitigation, fine-tuning strategy, or adversarial training.

### P4—Measurement Errors

Measurement errors refer to discrepancies and errors in the collected data. These errors can directly compromise data quality by creating spurious patterns and misleading information, leading to inaccurate predictions and thus contribute to the generalization gap. Three main types of such errors are:

**Labeling mistakes:** Different interpretations, subjective judgments, or inconsistencies in the labeling guidelines can cause human or automatic annotators to introduce errors while labeling the data. In fact, pervasive label errors have been identified in well-known datasets, with studies showing that around 3.3% of labels across 10 commonly-used datasets are incorrect, including 6% in the ImageNet validation set (Northcutt et al., 2021). Labels directly affect what the model learns. When the model bases its training on these incorrect labels, it will likely have incorrect predictions when facing real-world, unseen data, leading to a generalization gap. Therefore, careful annotation and quality control measures in the labeling process are crucial.

Additionally, when multiple annotators label the same data, disagreements may arise, and the strategy used to combine these labels is important. Simple majority voting is one approach, but more sophisticated methods, such as that proposed by Dawid & Skene (1979), which estimates individual annotator error rates, or CROWDLAB (Goh et al., 2022), which combines model predictions with annotations, can provide more accurate consensus labels and help reduce the impact of labeling errors on model performance.

**Noisy measurements:** This refers to the presence of noise or other unwanted artifacts in the measurements that can distort the learning process. These artifacts or noise can sometimes act as confounders, such as watermarks in image data, or medical interventions in medical data. Such artifacts are usually not intended to be part of the dataset but get included due to oversights in the measurement process. Oakden-Rayner et al. (2020) give an example of the presence of chest drain in a chest X-ray dataset. During error auditing by a board-certified radiologist, they discovered that the model learns to associate the visibility of a chest drain, a treatment tool, with pneumothorax, biasing predictions.

**Inappropriate proxies:** Features and labels are measured as proxies to quantify the concepts we deal with in ML. These proxies are simpler, measurable representations that substitute the more complex concepts. For example, for a model predicting house prices, features like "number of bedrooms" and "location" act as proxies to represent the concept of "housing value". Inappropriate proxies are the ones that are related to the target concept but are not truly representative of it, reflecting an error in the choice of what is being measured. One type of such proxies are simplistic proxies that are too simple for the desired concept and lead to inadequate models. For example, while age is a known risk factor for heart disease, predicting heart disease without also considering other influential factors like genetics, lifestyle, and other health conditions, makes it a simplistic proxy since it does not encapsulate the dynamics among various contributing factors. Some proxies may be inappropriate in the sense that there may be different mappings from the input proxy to the output proxy for different minority subgroups of the dataset. Suresh & Guttag (2021) gave an example of using "arrest history" as an input feature for risk assessment in the criminal justice system. Minority communities are often more likely to be arrested, because of different biases. Mapping from this proxy to the "riskiness" would thus differ for different subgroups in the data, which would lead to biased outcomes. Another important example of inappropriate proxies can occur due, for instance, to insufficient sampling rates where an image resolution does not match the scale of the features being investigated. For instance, attempting to identify micron-scale features in an image captured at millimeter resolution, or looking for millimeter-scale features in centimeter resolution images. This will likely lead to inadequate model performance. (Sabottke & Spieler, 2020).

To avoid these issues, it is important to carefully define and understand the data and the features, and this can again be achieved by incorporating domain knowledge. Careful labeling backed by rigorous guidelines for how to annotate the data is also crucial. There are ML methods that account for measurement and label uncertainties, but these techniques are not commonly used in practice. They may however be able to cope with potential inaccuracies or variability in the data. For example, label noise models actively acknowledge the potential for label errors during training (Song et al., 2022).

### P5—Historical Biases

Bias in data can stem from various sources, a few of which were discussed in the Dataset Shift pitfall (P2). For the other sources of biases in different stages of ML the reader can refer to Mehrabi et al. (2021). The pitfall of Historical Biases specifically highlights a type of bias that happens when existing stereotypical biases and prejudices creep into the sampled data. An example of this type of bias is mentioned by Suresh & Guttag (2021) where in a 2018 image search for women CEOs, resulted in fewer female CEO images due to the fact that only 5% of Fortune 500 CEOs were women at the time. The racial and gender disparities observed in the outputs of the Stable Diffusion model, as mentioned in the introduction, stem from a similar origin. Another example is the Correctional Offender Management Profiling for Alternative Sanctions (COMPAS) software used by the courts in the United States that measures the risk of a person to recommit another crime. An investigation into the software found a bias against African-Americans (Mehrabi et al., 2021). Note that this type of bias may happen even with perfect sampling, and actually most of the times it reflects the reality of our world. However, the question here is whether this reality that stems from human prejudice should be reflected by ML models (Suresh & Guttag, 2021). Moreover, as discussed earlier, ML models extract correlations between the inputs and outputs even when there are no real correlations (Geirhos et al., 2020). Therefore, they can make ethically dubious questions seem answerable. For example, the existence of numerous large face image datasets and sophisticated ML models for tasks on these images may legitimize attempting to predict subjective personal characteristics such as sexuality or political views solely based on face images, presuming that such correlations exist (Paullada et al., 2021; Schwartz et al., 2022). In-depth discussions on biases, their impact, and mitigation strategies in ML can be found in Barocas et al. (2017); Bender et al. (2021); Schwartz et al. (2022).

## 2.2 Model Design

The second step of the described ML pipeline, is to design and train a model for the specific problem at hand. This section focuses on three common pitfalls encountered in this phase.

### P6—Information Leakage

Information leakage refers to the exposure of information from test data into the training process. It is also called data leakage, data snooping, *etc.* That is, the training process is using some information that would not be available in practice. Information leakage can happen in various forms, that inflate the performance measures of a model and make us believe that it generalizes better than it actually does. Kapoor & Narayanan (2023) count leakage as an important reason behind the *reproducibility crisis* and reviewed 294 papers from 17 fields affected by leakage. A practical example of leakage is in the ICML 2013 Whale Challenge, where some participants discovered that specific characteristics of the dataset, such as patterns in file sizes and embedded timestamps, leaked information useful for the predictions. This gave their models an artificially high estimated generalization performance (Christopher, 2013). For an unbiased estimate of a model's generalization performance, the training, validation, and test sets must all be independent of each other and identically distributed. Furthermore, in many fields, for instance in healthcare, models are required to perform robustly across a variety of real-world conditions. It is increasingly being recognized that test sets should contain data from multiple real-world deployment sites, such as from several hospitals.

Although the necessity for having an independent test set may seem obvious, there are various common practices that lead to violations of this requirement, and some of them can be difficult to identify. Some of the common sources of information leakage are:

**Pre-processing:** Certain data pre-processing and transformations may involve using statistics computed from multiple data points. For example, normalization and scaling steps may require using the mean or standard deviations of multiple data points. It is crucial to be aware that such statistics should only be computed from the training data, and that those statistics computed from the training data should subsequently be used when normalizing the validation and test data too (Kapoor & Narayanan, 2023). If instead those statistics were computed on the entire dataset, including the validation and test data, then information from the validation and test set would *leak* knowledge into the training set. This is also something to consider when using resampling procedures, such as *e.g.*, the Bootstrap or cross-validation. In cross-validation, where data is divided into multiple subsets, any normalization or scaling must be done based on statistics computed only from the training folds, and not include data from the corresponding validation folds. Normalization can thus not be performed before the cross-validation, but must be performed within the cross-validation.

**Feature selection and extraction:** Similar to pre-processing, feature selection and extraction often involves using information from two or more data points. To avoid exposing information from the validation and test data, feature selection and extraction must also be performed only using the training set.

**Data slicing:** When processing for instance 3D data, time series, or video data, a common practice is to slice them into manageable subsegments. In the case of 3D data, this often involves slicing the information along one of the axes, turning volumetric data into a series of 2D slices. Similarly, time-series data or videos may be segmented into individual or groups of time points or frames, respectively. These individual segments may then be treated as separate, independent data samples that are fed to a model. However, this practice is prone to information leakage if the inherent dependencies within these segments are overlooked. For example, with volumetric MRI images and videos, slicing the data before splitting the dataset can cause adjacent slices or frames that are highly dependent to end up in both train and test data and by that violate the independence assumption.

Additionally, when dealing with time-series data, traditional cross-validation methods must be adapted to avoid information leakage. Specialized techniques, like time series cross-validation (Bergmeir & Benítez, 2012), prevent future data from inadvertently being used in the training process, ensuring the integrity of the forecasting models.

**Data augmentation:** Data augmentation is the process of creating new data points from existing ones, often through techniques such as rotation, scaling, and flipping, to artificially expand the dataset. This approach can be highly effective in improving model performance by enhancing the diversity of the training data and mitigating issues related to data sparsity (Xu et al., 2016). It is important to note that, this process creates dependencies between data samples. If the data set is augmented before splitting it up in training and test data, we may end up having dependent samples in train and test set, violating the independency requirement on the test set.

**Data over- or under-sampling:** Data over- or under-sampling are techniques used to address class imbalance problems (He & Garcia, 2009). Over-sampling involves synthesizing new samples for the minority classes, using the existing samples, to balance the dataset. Under-sampling involves removing samples from the majority classes, sometimes using information from the other samples when selecting samples to remove (Mani & Zhang, 2003). Similar to data augmentation, over- and under-sampling creates dependencies between samples, and it should not be performed on the entire data before splitting (Vandewiele et al., 2021). It must be performed on the training data after splitting up into training, validation, and test sets.

**Re-sampling:** Re-sampling is a technique where multiple subsets of the original data are created and used in an iterative process for training and validating the model to obtain more reliable performance estimates (Raschka, 2018). Common examples of re-sampling techniques include the bootstrap, jackknife, multiple variants of cross-validation (such as $k$-fold and leave-one-out), and permutation tests. Typically, at each iteration of a re-sampling procedure, a different subset of the data is used for validation and so there is a risk of information leakage when the data are split. For example, $k$-fold Cross-validation involves splitting the data into $k$ subsets, or folds, training the model on $k-1$ of the folds and validating on the remaining fold. This process is repeated multiple times (typically $k$ times), each time using a different fold as the validation set. If any of the previously mentioned steps, *i.e.*, pre-processing, feature selection, data augmentation, and over-sampling are performed outside the cross-validation loop, information from the validation fold may leak into the training process as discussed in Chapter 7.10.2 of Hastie et al. (2009). Therefore, all of the mentioned steps must be performed on the training data within the re-sampling to avoid information leakage.

**Hyper-parameter tuning:** Hyper-parameters typically have a substantial influence on the performance of a model and require careful tuning to ensure robust performance. However, hyper-parameter tuning, or optimization, can not involve the test set in any way. This is a recurring pitfall, and tuning hyper-parameters on the test set leads to a model that has adjusted to the characteristics of the test data, compromising the test set's role as an independent means to evaluate the model. Therefore, It is very important to view the test results only once, strictly for the final evaluation of the model, and have a separate validation set to tune hyper-parameters.

**P7—Model–Problem Mismatch**

The *no free lunch theorems* in ML states, roughly, that no single model excels at every task (Wolpert, 1996). The efficiency of a model varies across different domains, making model selection a crucial step when designing a ML system. *Occam's Razor* can serve as a guiding principle here, suggesting that simpler models should be favored over more complex ones if they achieve similar performance levels (Ghosh & Motani, 2021). This helps to avoid unnecessary complexity and reduces the risk that models over-fit specific datasets. The model selection process happens at two levels: choosing among different model types (like logistic regression, Support Vector Machine (SVM), K-Nearest Neighbor (KNN), *etc.*) and selecting within a model type based on different hyperparameters (for example, using different kernels in an SVM or different K values in KNN) (Murphy, 2012). In this pitfall we focus on the first level of model selection where we choose from different types of models. By "model-problem mismatch" we refer to situations when the chosen type of model does not align well with the specific requirements or characteristics of the problem at hand. Key issues in this pitfall include:

**Over-complicated/simplistic models:** The current trend in ML, driven by the availability of sophisticated methods and powerful computational resources, often encourages the belief that larger, more

complex models are essential for success. However, this belief can lead to using unnecessarily complex models when simpler ones might suffice (Fujinuma et al., 2022). Despite this trend in the research community, simpler models are often more efficient in practice (Paleyes et al., 2022). A relevant example is the experience of Haldar et al. (2019) with Airbnb's search algorithm, where a shift from a complex deep learning model to a simpler single-layer neural network not only improved manageability but also maintained good performance (Paleyes et al., 2022). Conversely, overly simplistic models may not sufficiently capture the underlying data patterns in some tasks, resulting in poor performance. For example, for image recognition tasks, linear models may fail to capture the intricate patterns in the data.

**Computational challenges:** Different models have varying computational requirements and scale differently. Choosing a model that is computationally expensive or that does not scale to large datasets may result in unnecessarily long training times, inefficiency, or infeasibility of deployment in real-world applications (Paleyes et al., 2022). This becomes particularly problematic when deploying large models on resource-constrained devices like internet of things/embedded devices or mobile devices, where it can become a problem to fit a large model on the device's memory or where too large models may lead to significant battery drain.

**Interpretability:** Model interpretability is crucial in many problem domains, particularly where the decision-making process needs clarity. While complex models like deep neural networks offer high performance, they often lack transparency in their decision making process. In high-stakes domains such as medical applications or in automobile/aviation settings, interpretability becomes a key model selection factor (Paleyes et al., 2022). In these contexts, choosing a simpler, more interpretable models such as linear models or decision trees can be more appropriate (Hansson et al., 2016; Rudin & Radin, 2019). Moreover, the 2018 explainable ML challenge showed that even in complex datasets, interpretable models could achieve similar accuracy to black box models, challenging the belief that complexity is necessary for accuracy (Rudin & Radin, 2019).

To avoid these issues, it is crucial to carefully analyze the problem, understand the characteristics of the data through extensive data exploration, and consider the strengths and weaknesses of different models. Engaging in effective communication with stakeholders is also essential, as many mismatches in ML can often be traced back to information gaps that could have been bridged through timely and clear discussions (Lewis et al., 2021). In general, choosing the best model involves considering the trade-offs between various factors such as performance, computational efficiency, interpretability, availability of data, and generalization ability of the model.

### P8—Over-/Under-fitting

Once the type of model is selected, we advance to the second level of model selection. This involves fine-tuning within a chosen model type through hyperparameter optimization. At this stage, the model is trained to learn the underlying patterns in the data, a process that essentially boils down to finding the best set of model parameters to minimize a loss function. In fact, training a model is all about maintaining the balance between a model's ability to capture complex patterns (low bias) and its robustness to noise or random fluctuations in the data (low variance). Once this balance is disturbed, the model's performance may suffer from overfitting or underfitting. Overfitting occurs when the model becomes too complex and starts fitting noise or irrelevant patterns in the training data, leading to poor generalization. Underfitting, on the other hand, happens when the model does not have high enough capacity, and it fails to capture the underlying complexity of the data. Both scenarios result in decreased performance and limit the model's ability to effectively handle new, unseen data.

Regularization is the usual way to control overfitting. A commonly used approach is to treat the capacity of a model as a hyperparameter, start with a low-capacity model (underfitted), and gradually increase the capacity until the point where the model starts to overfit to the training data. By tuning such hyperparameters on validation data, we can find the right level of complexity that minimizes overfitting while still capturing the essential patterns in the data. However, it should be noted that too much hyperparameter

tuning can lead to overfitting the validation set (Ng et al., 1997), a problem related to the concept of *p*-hacking in statistics (Head et al., 2015). This is when data or significance tests are collected or performed until a positive result is attained. This leads to a bias towards positive results (well-performing models in terms of the validation set in ML), and will thus increase the generalization gap (Simonsohn et al., 2014).

The above-mentioned concept, commonly referred to as the bias-variance trade-off has long guided the best practices in hyperparameter tuning of ML models. However, recent studies defy this conventional wisdom. Studies on a phenomenon known as "double descent" (Belkin et al., 2019) have revealed that in some cases after the point of interpolation, where a model perfectly fits the training data, the generalization error doesn't necessarily increase with model complexity. In fact, there is a second descent, where the generalization error decreases with the increase in the complexity, typically going below the previously considered optimal trade-off point. Recent work has shown that particular regularization can mitigate or even eliminate the double descent phenomenon in certain cases, leading to more stable generalization behavior as complexity increases (Nakkiran et al., 2020). There are also studies (Curth et al., 2024) that suggest that double descent can be reinterpreted as an artifact of different complexity measures. They show that when complexity is decomposed into separate axes, the apparent double descent folds back into a traditional U-shape. This insight reveals that the second descent may arise from shifts in the type of complexity rather than an inherent property of overparameterization.

Another interesting phenomenon known as "grokking" has also been recently observed (Power et al., 2022), where long after seemingly overfitting, the validation performance suddenly begins to improve from chance levels toward good generalization. Overall, controlling overfitting through hyperparameter tuning is one of the key factors in building robust ML models that generalize well, but it may be that the conventional wisdom need to be revised in the near future.

## 2.3 Model Evaluation

The third step of the described ML pipeline, is model evaluation. This section focuses on three common pitfalls encountered in this phase.

### P9—Misused Metrics

Metrics play a crucial role in evaluating the performance of ML models, providing quantitative measures of their performance. In fact, ML in its essence is the process of optimizing a metric with respect to some data. This makes Goodhart's Law a relevant pitfall, which states that "when a measure becomes a target, it ceases to be a good measure" (Goodhart, 2015; Thomas & Uminsky, 2022). This problem manifests itself in the following forms:

**Poor metric selection:** For choosing suitable metrics, it's important to consider the specific needs of the task and the characteristics of the data. For instance, in an image segmentation task where getting exact boundaries is important, relying on overlap-based metrics like the Dice Similarity Coefficient (DSC) might not be ideal, as they don't account for the accuracy of object boundaries (Reinke et al., 2023). Similarly, for datasets with imbalanced labels, we might achieve high scores on common metrics like accuracy while failing to detect a significant number of false negatives. For example, in a dataset for disease diagnosis, there might be 90% negative and only 10% positive cases, which can lead to models that are overly tuned to the majority class, resulting in a high number of false negatives for the rare, but important, positive cases. Another example is the different behaviors of the Root Mean Squared Error (RMSE) and Mean Absolute Error (MAE) depending on whether the error distribution is normal or Laplacian (Hodson, 2022).

**Poor implementation:** Once selected, the implementation of a metrics can also be prone to various pitfalls. Metrics implementation is not an standardized process, and the choices you make can change the resulting value (Varoquaux & Cheplygina, 2022). For example, Reinke et al. (2023) illustrate that for the Average Precision (AP) metric and the construction of the Precision-Recall (PR)-curve, the strategy used for handling identical scores has a notable impact on the final metric values.

**Gaming:** When a metric is given too much importance, the risk of gaming that metric becomes an inevitable concern. As we strive to achieve favorable outcomes, there will exist a temptation to optimize solely for the chosen metric rather than genuinely enhancing the underlying system's performance (Thomas & Uminsky, 2022). An example of this can be seen in modern recommendation systems used on various platforms. These systems are often manipulated by tactics like fake clicks, reviews, and followers to artificially increase content rankings and visibility (Tufekci, 2019; Thomas & Uminsky, 2022). Search Engine Optimization (SEO) is another example which aims at improving a website's ranking in search results using metrics that not all are related to the true relevance and quality of the site. Note that the very choice of a metric can also be subjected to gaming. Metrics showing favorable performance can be chosen, while those reflecting a model's true generalizability are overlooked.

**Short-term concerns:** Metrics are only proxies for what we really care about, and relying solely on them for model evaluation can overlook other critical factors, such as interpretability, fairness, ethical considerations, and our impact on society and environment. For instance, in news media, a controversial opinion piece by an influencer may obtain higher click-through rates compared to a thoughtful, in-depth article by an expert. When decision-making revolves exclusively around quantifiable measures, the scope of evaluation narrows down to short-term gains or outcomes that are easily observable and quantified. This means complex, nuanced, or long-term factors will be overlooked or undervalued (Thomas & Uminsky, 2022). For instance, Bernardi et al. (2019) report that in the recommendation system of Booking.com, optimizing for click-through rate (CTR) without accounting for user experience can lead to the "paradox of choice," where users are overwhelmed with highly similar recommendations that may increase clicks but ultimately harm satisfaction and conversion.

To deal with these problems one important strategy is to use diverse types of metrics that collectively capture various facets of a model's performance producing a fuller picture. Moreover, it reduces the potential for gaming, as no single metric becomes the sole focus, and it is more complex to game several metrics at once (Reinke et al., 2023). Additionally, integrating qualitative metrics, such as user satisfaction surveys and expert assessments, provides a richer perspective on a model's real-world impact beyond numerical outputs (Thomas & Uminsky, 2022; Reinke et al., 2023).

### P10—Black Box Models

Black box models refer to ML methods that produce predictions or decisions without providing a clear explanation of how they arrived at those results. Neural networks, SVMs, and ensemble classifiers are common examples of such methods (Guidotti et al., 2018). While powerful, their lack of interpretability makes the evaluation of such methods a challenging task. Without transparency into the decision making process it becomes difficult to identify and fix any biases, errors, or failure modes that lead to reduced generalization performance.

Using black box models can compromise trust and accountability, particularly in high-stake domains where clear explanations are essential, such as healthcare or finance. In some cases, the use of a black box model may not even be necessary. For example, during the 2018 Explainable Machine Learning Challenge, participants were asked to design black box models for loan default prediction, which is a common task in finance. However, a team showed that an interpretable model could effectively predict loan defaults, with the same accuracy as more complex black box models (Rudin & Radin, 2019). Note that explainability is particularly important in this task, where lenders and borrowers can benefit from understanding the factors influencing credit decisions, and therefore, ensuring fairness and meeting regulations.

When black box models are used, there are post-hoc Explainable AI (XAI) methods available to try to explain their decisions. Linardatos et al. (2020) discuss two main types of these methods. First, there are general methods such as Local Interpretable Model-Agnostic Explanations (LIME) (Ribeiro et al., 2016) and SHapley Additive exPlanations (SHAP) (Lundberg & Lee, 2017) that are applicable to any black-box model. Second, there are methods specifically designed for deep neural networks and image analysis, including saliency-based (Itti et al., 1998) and gradient-based (Simonyan et al., 2019) approaches.

As the complexity and opacity of deep learning models increase, the role of XAI methods becomes important. These methods are not only essential for transparency but also can help recognizing and addressing many of the pitfalls associated with ML models. For example, XAI can be helpful in identifying confounders that influence outputs in unexpected ways. It can detect biases embedded within data or model behaviors. Moreover, it helps diagnose overfitting by revealing whether decisions are based on noise rather than relevant features.

**P11—Baseline Comparisons Issues**

To benchmark, to compare, against established models or methods and/on well-known sets of data is a critical step in evaluating a developed model. There are pitfalls that we may encounter when doing this comparison. These pitfalls can contribute to a generalization gap, by making the model appear more effective in test environments than it actually is in real-world applications. Two important ones are:

**Unfair comparisons:** Without fair comparison, any drawn conclusions may be questionable. This includes ensuring that all models undergo similar pre-processing and equal hyperparameter tuning. The latter is particularly important, because according to our review results presented in Question 11 of Figure 7, it is often overlooked. If you tune hyperparameters for your proposed model but use default or suboptimal hyperparameters for the baselines, the comparison might not be fair. Therefore, the recommended practice is to employ a validation set and to fine-tune the hyperparameters of all the compared models to that validation set (Lones, 2021). In addition, if possible, it is important to use similar hyperparameter tuning methods and budgets for all the compared models. For instance, using grid search for one model and Bayesian optimization for another, or ten hyper-parameter search rounds for one model but 100 for another, will increase the risk of the comparison being biased in favor of the latter option in both cases.

**Ignoring uncertainties:** Despite the importance of uncertainties in understanding model behavior and decision-making, most ML approaches and articles do not provide a real estimate of their predictions' uncertainties. When we look at the experimental results in papers, we often see tables displaying the *best* results, highlighted with a bold single number. However, it is important to note that the output of a model is a random variable, which in practice means that whatever value we happen to get from it is uncertain. These uncertainties are often overlooked when comparing different methods as suggested by the Question 12 results of Figure 7 in our review. To get a clearer picture of the model performance and limitations, and to make reliable decisions, we need to account for these uncertainties. Uncertainties in ML can be categorized into aleatoric and epistemic sources. Aleatoric uncertainties are the result of inherent randomness in the collected data, and it cannot be reduced by collecting more of the same data. Techniques such as data augmentation may be able to mitigate some aleatoric uncertainties. Epistemic uncertainty, on the other hand, comes from the imperfections in the model itself and us as analysts, such as our ignorance of the best model or about the data generating process, or that found model parameters are suboptimal. Epistemic uncertainty is the results of a lack of knowledge, often due to limited amounts of data. Epistemic uncertainties can be addressed by gathering more data and can be quantified using Bayesian approaches (Kendall & Gal, 2017; Hüllermeier & Waegeman, 2021).

Three important points to keep in mind are as follows. First, a single run of a ML method is affected by various random factors, such as *e.g.*, the initial (random) model parameter values and the particular data set we have (Pham et al., 2020). We can estimate the uncertainty in the predictions by *e.g.*, training and applying the method multiple times using resampling techniques such as cross-validation or the Bootstrap (Efron, 1992), or uncertainty quantification methods such as conformal prediction (Gammerman et al., 2013; Shafer & Vovk, 2008). These methods provide a more stable comparison of the baselines, when we use their confidence, credible, or prediction intervals to compare results, since they account for the uncertainties present. Second, uncertainties in performance metrics can be evaluated by computing and reporting quantities such as standard error, confidence intervals, or other dispersion metrics that encapsulate the range of possible outcomes. Third, even if one model outperforms another in terms of the mean metric computed, the differences might not be

statistically significant. Several statistical tests can assess whether differences in model performance are significant. According to Demšar (2006), when comparing two classifiers, the Wilcoxon signed-rank test is suitable for non-normal data distributions, while the paired t-test is commonly used for normally distributed results. For comparing multiple classifiers, the Friedman test is preferred when the assumptions of parametric tests such as normality and homoscedasticity are violated, and ANOVA can be applied when these assumptions hold. For more detailed discussions on these statistical methods and their applications in classifier comparisons, refer to Demšar (2006).

Furthermore, it is important to evaluate a model across multiple datasets to validate its performance under different conditions. Equally important is to avoid cherry-picking where favorable outcomes are selectively reported, and transparently publish all results, not just the favorable ones. This practice helps to ensure that the community gains a realistic understanding of a model's capabilities across a variety of scenarios, rather than a skewed view that could lead to misleading conclusions about its efficacy and applicability.

## 2.4 Reporting

The fourth and final step of the described ML pipeline, is report. This section focuses on three common pitfalls encountered in this phase.

### P12—Insufficient Reports

ML studies often fall short in providing clear, detailed reports (Chiarelli et al., 2021). Chiarelli et al. (2021) mention insufficient reports as one of the key barriers to reproducible publication practices. This issue, possibly due to page limits or a general underestimation of thorough reporting by researchers, is evident in our review findings. As discussed in Section 4, we observed noticeable lack of clarity while analyzing the reporting trends regarding the previously presented pitfalls in the papers from ICCV and MICCAI conferences. Without precise information about the details of the developments that were made, it will be very difficult for other researchers in the community to understand, validate, and replicate the study. Consequently, they can't verify or challenge the findings, and latent issues can go unnoticed. Such lack of clarity can inadvertently promote models that appear effective but have hidden biases, overfitting issues, or other problems. This can thus lead to a reduced generalization performance, and hence an enlarged generalization gap. There are several standards and guidelines that can be used to ensure a comprehensive report. For example, the CLAIM (Checklist for Artificial Intelligence in Medical Imaging) is a useful standard that provides specific guidelines for authors and reviewers working in the domain of AI application in medical imaging, and it is meaningful also in other domains (Mongan et al., 2020). Similarly, the paper checklist guidelines from NeurIPS provides a great resource to ensure that all vital components of the research are transparently reported (Pineau et al., 2021).

### P13—Faulty Interpretations of the Results

Another issue with ML report is the potential for erroneous interpretations of the model's performance. As mentioned earlier one model is never better than another in every aspect, and optimizing one aspect often means compromising another (Wolpert, 1996). Not adequately reflecting these trade-offs in conclusions creates a false impression of the model's effectiveness. Another common mistake is to draw general conclusions beyond the data used (Lones, 2021). If a model performs well on one dataset, it does not mean it will do the same on another (Paullada et al., 2021). A possible solution to this challenge would be if the community developed diverse benchmark datasets that are constructed to evaluate various aspects of model performance. This would ensure comprehensive assessment across different scenarios and would promote the development of models that generalize better.

It is also important to transparently report each model's strengths and limitations, as encouraged by conferences like NeurIPS. They advise the authors to create a separate "Limitations" section, discuss assumptions, robustness to these assumptions, and factor influencing the performance (Pineau et al., 2021). Such transparency not only can pave the way for future improvements but also guard against premature application of potentially flawed methods.

Another common issue in interpreting results of a ML model is the temptation to draw causal insights from the results (Molnar et al., 2020). For instance, in a medical application, ML models may predict how a patient might respond to a drug based on past data. But these predictions are often just correlations, not actual causes. True *actionable* insights, necessary for personalized treatments, require understanding the causal effects of interventions, which is something ML models typically do not provide (Sanchez et al., 2022). It's important to also note that there may be a misconception that interpretability methods provide causal understanding. While traditional interpretable ML primarily focuses on the correlations rather than the causality (Xu et al., 2020), increasing attention is being devoted to causality-oriented interpretability methods (Kaddour et al., 2022; Zhao & Hastie, 2021).

### P14—Code Reproducibility Issues

Difficulties in reproducing the codes from ML research papers is another pitfall limiting the generalizability and practical application of these models in real-world scenarios. Haibe-Kains et al. (2020) argue that while sharing complex computational pipelines of a research is common in areas like genomics, in other fields of science data and code are not commonly made available. This trend is also visible in our review outcomes as shown by the Question 13 results of Figure 7.

The probability that a research paper has a real-world impact increases when the results can be validated and reproduced by others (Haibe-Kains et al., 2020). Although code sharing does not directly affect generalization, it enhances transparency, enabling the community to verify generalization claims across varied conditions, thereby increasing trust in the model's robustness. The development of a ML model involves a wide range of choices, theoretical details, and implementation details, and even with a very well-documented report, code sharing remains essential to publicly provide these detail (Gundersen & Kjensmo, 2018). A well-organized code repository can eliminate the ambiguity in interpreting the written description of a model. It can also save time by eliminating the time-consuming task of reproducing a method based on the textual description in a paper.

Simkó et al. (2024) proposed a checklist with eight essential items, based on previous such attempts in the ML community, with the purpose to ensure the reproducibility of shared code repositories. First, address all the publicly available materials in the main body of the paper. Second, if possible provide a public link to the code repository. Third, if possible make datasets publicly available; if not, evaluate your model also on a similar publicly available dataset, to make at least parts of the work reproducible. Fourth, clearly list and describe all dependencies and their version numbers. In this regard, it is important to mention the framework used, such as PyTorch and TensorFlow, their version numbers, and other implementation details such as constants used in the study. Fifth, provide the code to build and train your model with the exact hyperparameters used. Sixth, provide the code to evaluate your model with the metrics presented in the paper. Seventh, share the trained model, including trained weights. Finally, provide documentation (such as a readme file) with a detailed description of the repository to help the reader to understand the code and how to use it.

While code and data sharing improve reproducibility, several challenges can impede these practices. Authors may hesitate to share their work due to concerns over intellectual property, privacy issues, or the substantial effort required. Furthermore, current academic incentives often prioritize novel findings over reproducibility. To address these issues, journals, conferences, and funding agencies could provide better recognition and support for efforts that promote transparency and reproducibility, therefore encouraging more researchers to adopt these practices.

## 3 Experiments

In this section, we aim to empirically illustrate the impact of three selected pitfalls, namely dataset shift (P2), confounders (P3), and data leakage (P6), using the Brain Tumor Segmentation (BraTS) dataset (Baid et al., 2021; Menze et al., 2014). This is a popular dataset for benchmarking ML models in the context of medical imaging, specifically focusing on semantic segmentation of brain tumors from multimodal magnetic resonance imaging (MRI) scans. Through a series of simple experiments, we will quantitatively investigate how each

identified pitfall can skew the perceived performance of ML models, leading to erroneous conclusions and compromised generalization.

**Experiment 1—Dataset Shift**

As mentioned earlier, dataset shift occurs when the distribution of the data the model encounters post-deployment differs from the data it was trained on. In this experiment, we study this pitfall by simulating conditions that reflect real-world disparities in MRI data acquisition across different medical centers, *i.e.*, non-stationary environments. The BraTS dataset, containing MRI scans from various medical sites, allows us to utilize site information to construct subsets of this data for a controlled study of this pitfall.

**Data and task:** For this experiment we considered the BraTS 2023 Adult Glioma segmentation dataset[1]. This dataset contains 5,880 MRI scans across four different modalities from 1,470 brain diffuse glioma patients. To design a simple segmentation task, we focus on slice-level segmentation where each transversal 2D slice in the axial direction is treated as an independent data point and serve as the input for our ML model. The model is designed to take all four MRI modalities available in the BraTS dataset—T1-weighted, T1-weighted with contrast enhancement (T1ce), T2-weighted, and T2-weighted Fluid Attenuated Inversion Recovery (T2-FLAIR)—as a combined input, resulting in four input channels. Moreover, rather than distinguishing between the three different tumor sub-regions originally annotated in the dataset, we merged them into a single tumor region, thus converting the task into a simpler binary segmentation problem.

To simulate real-world dataset shift, we used the site number information (representing the originating institution for each MRI scan) to construct two distinct subsets of the data, presented below as Case 1 and Case 2. Case 1 represents a scenario where we do not have a shift from train to test datasets, while in Case 2 we do have a shift.

**Case 1:** Focused on a single-site development scenario, we construct the train, validation, and test sets as follows:

>*Train and validation*: We randomly selected 30 patients from one of the sites, namely Site 1, for training and another 10 patients from the same site for validation.

>*Test*: Two test sets were constructed: "test" containing 10 patients from the same site as train and validation (Site 1), and "ex-test" containing 10 patients from a different site (Site 18). The sites were chosen randomly and based on the availability of data.

**Case 2:** Focused on a multi-site development scenario, we construct the train, validation, and test sets as follows:

>*Train and validation*: We randomly selected 30 patients from six different sites, namely Sites 1, 4, 13, 21, 6, and 20, for training and another 10 patients from the same sites for validation.

>*Test*: We used the exact same "test" and "ex-test" as in Case 1.

Note that in both cases, "ex-test" serves as an independent (external) test set, representing data coming from a new medical center not seen during the training and validation. The difference between the performance on test and ex-test sets is what we call the generalization gap, and it is what this experiment is focused on.

After dividing the dataset into training, validation, and two test subsets according to the presented cases, 2D slices were extracted from the MRI scans. Originally, each scan contained a volume image with resolution $240 \times 240 \times 155$. From this, we focused on the central region, and extracted one slice out of every three from the 100 middle slices, which resulted in 34 slices per patient. Note that, by extracting slices only from the center, we ensured that the resulting 34 slices for each patient contained segmentation masks. Therefore, no slices without segmentation masks (where all elements would be zero) were included in the data. This provided us with around 3,000 slices for training, and 1000 slices each for validation, test, and ex-test. Each slice was then normalized by scaling the pixel values between 0 and 1 using min-max normalization and

---

[1]The dataset can be accessed at the Synapse platform: `https://www.synapse.org/#!Synapse:syn51156910/wiki/621282`

resized to $128 \times 128$ pixels. Note that to ensure that the results do not depend on the specific subset of the BraTS data that we select, we repeat the whole explained data extraction procedure 25 times with different random seeds to construct 25 distinct dataset as detailed above. That is, each of these 25 datasets contain the two cases and we compute the performance on each dataset, and the results are averaged over these 25 datasets.

**Model design:** A standard U-Net architecture was used for this segmentation task, similar to the one described in the original U-Net paper (Ronneberger et al., 2015), with an initial filter size of 64. As mentioned the model takes all four modalities of MRI scans as a combined input, resulting in four input "channels". The output of the model was a binary mask indicating tumor presence in each pixel. The Adam optimizer (Kingma & Ba, 2014) was used for training. The learning rate was set to $5 \cdot 10^{-5}$ after a grid search over $\{5 \cdot 10^{-3}, 1 \cdot 10^{-3}, 5 \cdot 10^{-4}, 1 \cdot 10^{-4}, 5 \cdot 10^{-5}, 1 \cdot 10^{-5}, 5 \cdot 10^{-6}, 1 \cdot 10^{-6}, 5 \cdot 10^{-7}, 1 \cdot 10^{-7}\}$, which yielded the best validation accuracy. The models were trained for up to 100 epochs, with early stopping applied when the validation loss failed to improve for 10 consecutive epochs. Improvement was determined by any absolute decrease in validation loss relative to the best value observed so far.

To evaluate the segmentation performance, the same metrics as described by Vu et al. (2021) were employed: The DSC that is used to measure the overlap between the predicted and actual segmentation, with an optimal value of 1 indicating perfect agreement. The 95'th percentile Hausdorff Distance (HD95) where a value of 0 represents ideal boundary alignment between the predicted and actual segmentation. The Relative absolute volume difference (RAVD) that calculates the difference in volume (or area in 2D images) between the segmented and actual regions, normalized by the volume of the actual region, with 0 being the optimal value indicating exact volume match. The Average Symmetric Surface Distance (ASSD) that computes the average distance between the boundary points of the predicted and actual segmentation, where an optimal score of 0 represents perfect boundary alignment. A combined loss function, that integrated both the Dice loss and binary cross entropy, was used as the models' loss functions (Vu et al., 2021). The binary cross entropy was weighted at 0.3 and Dice loss at 0.7, with the 0.3 value for binary cross entropy chosen based on a grid search over the values $\{0, 0.1, 0.3, 0.5, 0.7, 0.9, 1\}$. This loss function was used to align with common practices in semantic segmentation tasks (Isensee et al., 2018). The additional metrics ensure a comprehensive evaluation.

**Results and discussion:** Figure 2 presents the segmentation results for both Case 1 (blue color) and Case 2 (red color) across the train, validation, test, and ex-test sets, containing results for all four metrics (DSC, HD95, RAVD, and ASSD). In these plots, the circles at the center of the error bars indicate the mean metric values, averaged over 25 datasets, and the error bars illustrate a 95% confidence interval. Table 2 details the mean values (across 25 datasets) and standard errors for the test and ex-test sets for each case along with the generalization gaps.

It can be observed in Figure 2 that, as expected, there is a drop in segmentation performance moving from train to validation and further to the test sets across most metrics (except RAVD). The drop in performance from validation to test set illustrates the importance of having a separate test set to obtain a realistic estimate of the model's performance. The drop in performance from test to ex-test set is what we call the generalization gap, and it can be observed in Figure 2 and Table 2. Note that, for the DSC metric, where higher values indicate better performance, the gap is calculated as Test - Ex-test; for HD95, RAVD, and ASSD, where lower values are preferable, the gap is calculated as Ex-test - Test. According to mean values in Table 2, the gap is more pronounced for Case 1, where the training was limited to data from a single site. In contrast, Case 2 exhibits a reduced generalization gap that can be attributed to the increased diversity in its training data. The wider dataset variety in Case 2 helps mitigate the dataset shift, demonstrating better generalization on an external test set.

However, as mentioned the above conclusion is made based on the mean values of the gaps observed in two cases. To investigate whether the difference in the generalization gaps between Case 1 and Case 2 is statistically significant we have performed paired t-tests on the difference between gaps of the two cases for each metric across the 25 datasets. The goal was to see if the generalization gaps (indicated as *Gap* in Table 2) observed in Case 1 were significantly larger than those observed in Case 2 for each metric. The results presented in Table 3 show p-values for the DSC, HD95, RAVD, and ASSD metrics. Figure 3 depicts

the distribution of the differences in gaps between Case 1 and Case 2 (*i.e.*, Gap of Case 1 - Gap of Case 2). The distributions are estimated using Kernel Density Estimation (KDE). The p-values of the t-tests show a statistically significant difference in the generalization gap measured by the DSC, HD95, and ASSD scores ($p = 1.31 \times 10^{-7}$, $p = 5.94 \times 10^{-4}$, and $p = 3.60 \times 10^{-3}$, respectively), but no statistical difference for the RAVD score ($p = 8.79 \times 10^{-2}$) according to the 0.05 significance threshold. To account for the fact that four different metrics were used, we Bonferroni corrected the significance threshold to $0.05/4 = 0.0125$. With this stricter threshold, the results for DSC, HD95, and ASSD remain statistically significant. It is important to note that, in constructing the 25 datasets with different random seeds, some degree of overlap did occur between the sampled datasets (about 10% overlap on average). This overlap introduces a dependency among datasets, affecting the strict independence assumption required for the paired t-test. Consequently, interpretation of these results should be approached with caution.

Each metric captures different aspect of the model's performance. The DSC score focuses on overlap accuracy, while the HD95 and ASSD scores focus on boundary delineation. The results show that for both Case 1 and Case 2, DSC, HD95, and ASSD scores generally indicate accurate performance, but they reveal a noticeable decline for the ex-test set, showing the challenge in generalizing to new, unseen data. Interestingly, the RAVD score, which measures the model's accuracy in estimating tumor volume, remains stable and close to zero across both cases. This suggests that while the models might struggle with precise boundary and shape delineation, they are reliable in estimating tumor volume. However, it's important to note that RAVD can show a perfect score of zero even in imperfect segmentation, as long as the segmented volume matches the ground truth volume. This variation in metric performance highlights the metrics pitfall (P9) and emphasizes the importance of using a combination of different metrics for evaluation.

Note that this experiment, utilizing the BraTS dataset, serves as a simplified illustration of the dataset shift phenomenon. Despite the uniform pre-processing and standardization across different sites within the BraTS dataset, we still observe a dataset shift from one medical center to another, which is more significantly observed by the DSC metric. This shift is expected to be even more severe in real-world scenarios where each data source may follow different pre-processing protocols, thus increasing the dataset shift challenge.

Table 2: Obtained segmentation results for Experiment 1 measured by DSC, HD95, RAVD, and ASSD scores. The table shows the mean metric values, averaged over 25 datasets, and the numbers in parentheses indicate the standard errors. Case 1 and Case 2 represent single-site and multi-site development scenarios, respectively. Ex-test serves as an independent (external) test set, and Gap represents the difference between test and ex-test results or the generalization gap. Note that, For the DSC metric, where higher values indicate better performance ($\uparrow$), the gap is calculated as Test - Ex-test; for HD95, RAVD, and ASSD, where lower values are preferable ($\downarrow$), the gap is calculated as Ex-test - Test.

| Metric | Case 1 | | | Case 2 | | |
|---|---|---|---|---|---|---|
| | Test | Ex-test | Gap | Test | Ex-test | Gap |
| DSC $^\uparrow$ | 0.82 ($\pm$0.01) | 0.69 ($\pm$0.01) | 0.13 | 0.80 ($\pm$0.01) | 0.75 ($\pm$0.01) | 0.05 |
| HD95 $^\downarrow$ | 1.99 ($\pm$0.13) | 4.79 ($\pm$0.39) | 2.80 | 2.13 ($\pm$0.15) | 3.60 ($\pm$0.35) | 1.47 |
| RAVD $^\downarrow$ | -0.05 ($\pm$0.02) | -0.01 ($\pm$0.04) | 0.04 | 0.03 ($\pm$0.06) | -0.01 ($\pm$0.03) | -0.04 |
| ASSD $^\downarrow$ | 0.44 ($\pm$0.05) | 1.46 ($\pm$0.18) | 1.02 | 0.52 ($\pm$0.08) | 1.05 ($\pm$0.16) | 0.53 |

Table 3: The paired t-test results for the difference in generalization gaps observed for Case 1 and Case 2. The tests were conducted on 25 datasets, with 24 degrees of freedom.

| | DSC | HD95 | RAVD | ASSD |
|---|---|---|---|---|
| T-Statistic | 7.06 | 3.67 | 1.39 | 2.93 |
| P-Value | $1.31 \times 10^{-7}$ | $5.94 \times 10^{-4}$ | $8.79 \times 10^{-2}$ | $3.60 \times 10^{-3}$ |

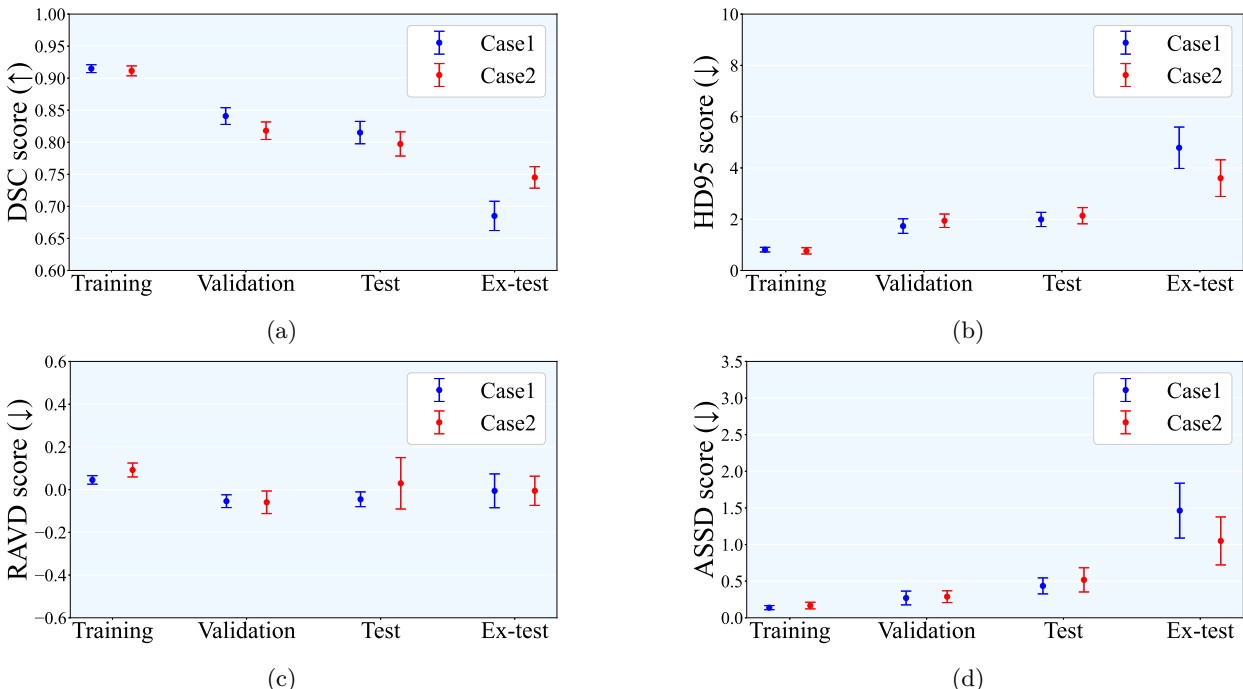

Figure 2: Obtained segmentation results for Experiment 1 measured by (a) DSC, (b) HD95, (c) RAVD and, (d) ASSD scores. The circles at the center of the error bars indicate the mean metric values, averaged over 25 datasets, and the error bars illustrate a 95% confidence interval. Case 1 (blue color) and Case 2 (red color) represent single-site and multi-site development scenarios, respectively. Ex-test serves as an independent (external) test set, and the difference between test and ex-test results represents generalization gap. Note that, For the DSC metric, higher values indicate better performance ($\uparrow$) and for HD95, RAVD, and ASSD lower values are preferable ($\downarrow$).

### Experiment 2: Spurious Correlations

The second experiment is focused on the spurious correlations pitfall (P3), where ML models tend to rely on shortcuts that do not generalize well across different data distributions.

**Data and task:** In this experiment, we use the BraTS 2021 Task 2 challenge (Baid et al., 2021), focusing on classifying the $O^6$-MethylGuanine-DNA MethylTransferase (MGMT) promoter methylation status in brain tumors from mpMRI scans. This status is a binary label defined as unmethylated (0) or methylated (1). We choose this dataset because there are studies indicating that various deep learning models face challenges in detecting a reliable correlation between MRI imaging data and the MGMT promoter methylation status, raising the question of whether such a correlation even exists (Kim et al., 2022; Saeed et al., 2023). Therefore, using this dataset, the aim was to show how shortcuts can easily bias the results in the absence of strong correlations between input and output.

To simplify the task to an image-level classification, we extracted every other slice from the central 100 slices of each patient's MRI scan, and excluded any slices with empty tumor masks. From the original dataset, we thus selected a subset of the data with 50 patients (25 from each class) for training, 20 (10 from each class) for validation, 20 (10 from each class) for testing, and another 20 (10 from each class) as an external test set. To study the effect of different shortcuts, this subset of the data was extracted in three different cases elaborated below. In this setup, we deliberately introduced sampling bias by sourcing Class 0 patients exclusively from a single medical center, creating conditions where the model could rely on spurious correlations arising from site-specific characteristics. Figure 4 illustrates how these three cases were designed. Note that the external test set (referred to as ex-test hereafter) was identical in all three cases and was free from shortcuts. Its

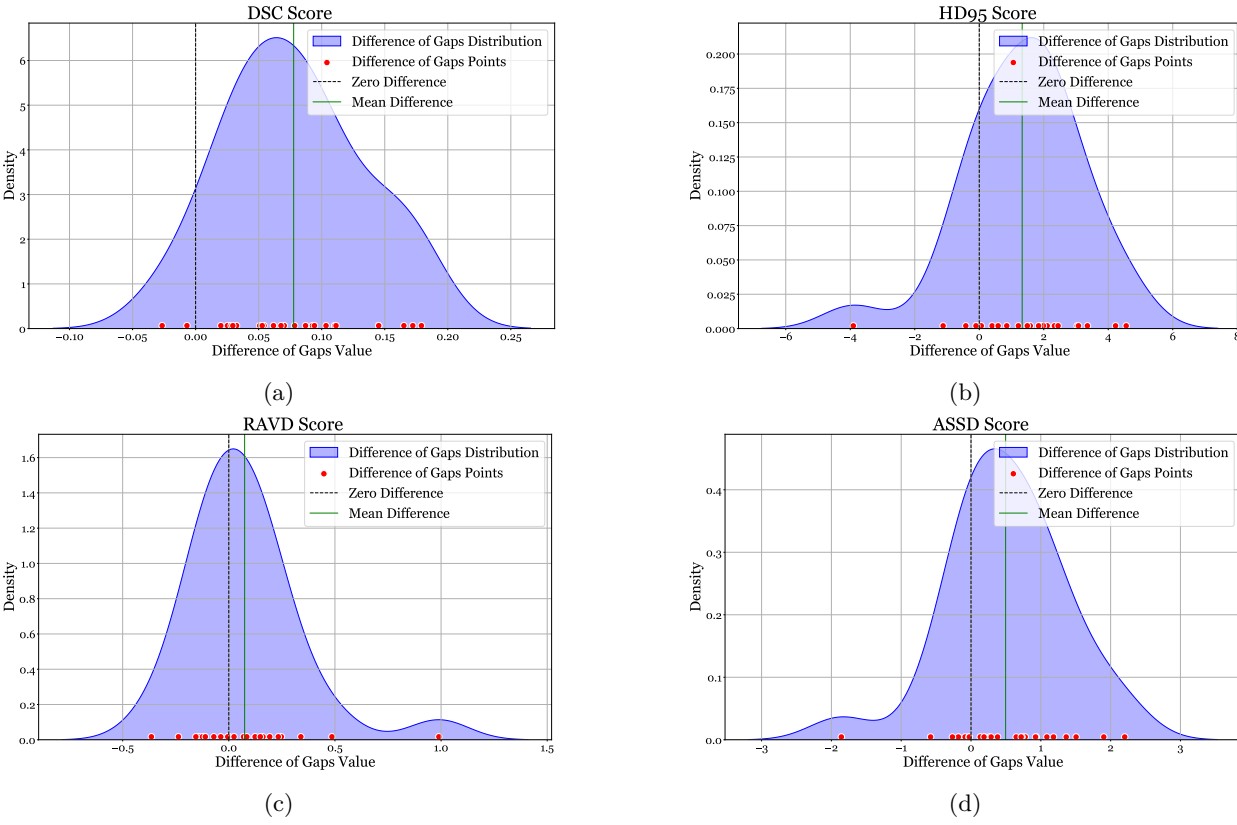

Figure 3: The distribution of the differences in generalization gaps of Case 1 and Case 2 in Experiment 1, where the "Difference of Gaps" variable in the figure represents the gap of Case 1 minus the gap of Case 2. The distributions are estimated using Kernel Density Estimation (KDE).

design was similar to the test set in Case 3 of this experiment described below, and it was used to evaluate the model's generalizability.

The three cases designed in this experiment are:

**Case 1:** In this case all the data from the unmethylated (0) class in train, validation, and test sets were from a single medical center, and their images were intentionally modified by adding subtle Gaussian noise, with a mean of zero and a standard deviation of 0.0006. This specific noise profile was designed to be visually imperceptible with the naked eye while simulating a scenario where data from one hospital have inherent specific noise characteristics.

**Case 2:** The dataset in this case was similar to Case 1 with all unmethylated (Class 0) patients originating from a single hospital. However, no Gaussian noise was added to these images. This setup allowed for the assessment of the impact of site-specific data characteristics without the effect of added noise.

**Case 3:** Serving as the control case, this dataset was constructed by randomly selecting patients, irrespective of their origin site, and without the addition of any noise, removing the potential shortcuts.

**Model design:** Choosing a model for this task is challenging, given the difficulties ML models face in detecting a reliable correlation between MRI imaging data and the MGMT promoter methylation status, as was discussed above. Since the goal of this experiment was to demonstrate the effects of confounders, and not to find the best model for this task, we used the same standard U-Net architecture as in Experiment 1 for simplicity. To adapt this model for classification, we augmented it with a classification block. This block

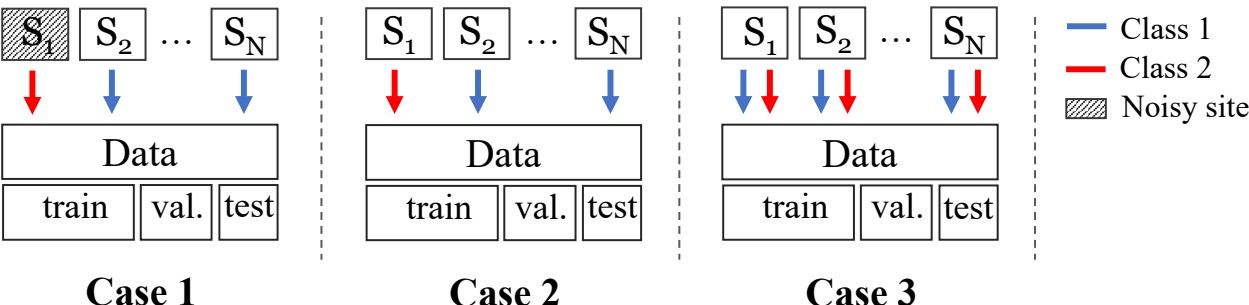

Figure 4: The illustration of the data extraction process for Experiment 2. The blue and red arrays represent data from class 1 and class 2, respectively. $S_1$, $S_2$,..., $S_N$ represent the origin site of the data. The hatched square indicates the site with added noise and "val." is short for validation data.

included a Flatten layer, two Dense layers with ReLU activations (256 and 32 neurons), a Dropout layer, and a final Dense layer with a sigmoid activation. The dropout rate was set to 0.5 after a search over values $\{0.4, 0.5, 0.6, 0.7\}$. The models were trained for 100 epochs, and the model with the lowest validation loss during training was selected for testing. The learning rate was set to $5 \cdot 10^{-5}$ as in Experiment 1.

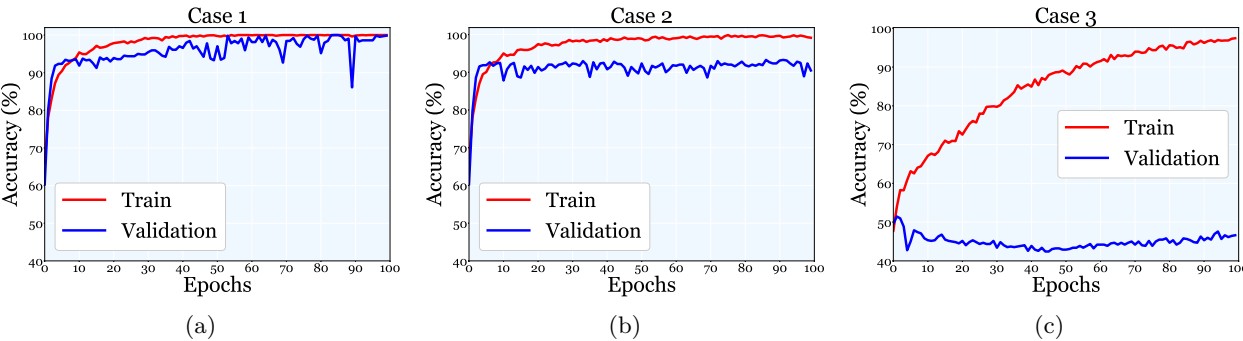

Figure 5: The obtained learning curves for the three cases in Experiment 2. (a) Case 1: data with noise as confounder. (b) Case 2: data without noise but with site information as confounder. (c) Case 3: random data without confounders.

Table 4: Classification results measured by accuracy, precision, recall, f1-score, and AUC metrics on test and external test sets for experiment 2. The table shows the mean metric values obtained through bootstrap evaluation with 1000 iterations, and the numbers in parentheses indicate the standard errors.

|  |  | Accuracy (%) | Precision (%) | Recall (%) | F1-score (%) | AUC (%) |
|---|---|---|---|---|---|---|
| Case 1 | Test | 99.66 ($\pm$0.000) | 99.32 ($\pm$0.000) | 100.00 ($\pm$0.000) | 99.65 ($\pm$0.000) | 100.00 ($\pm$0.000) |
|  | Ex-test | 51.70 ($\pm$0.001) | 51.70 ($\pm$0.001) | 100.00 ($\pm$0.000) | 68.14 ($\pm$0.001) | 48.99 ($\pm$0.001) |
| Case 2 | Test | 95.13 ($\pm$0.000) | 94.81 ($\pm$0.000) | 95.40 ($\pm$0.000) | 95.10 ($\pm$0.000) | 99.04 ($\pm$0.000) |
|  | Ex-test | 52.93 ($\pm$0.000) | 52.57 ($\pm$0.001) | 91.99 ($\pm$0.001) | 66.88 ($\pm$0.001) | 50.07 ($\pm$0.001) |
| Case 3 | Test | 43.10 ($\pm$0.001) | 42.09 ($\pm$0.001) | 22.58 ($\pm$0.001) | 29.36 ($\pm$0.001) | 43.48 ($\pm$0.001) |
|  | Ex-test | 62.95 ($\pm$0.001) | 74.16 ($\pm$0.001) | 38.25 ($\pm$0.001) | 50.41 ($\pm$0.001) | 67.03 ($\pm$0.001) |

**Results and discussion:** Figure 5 presents the learning curves for each of the three experimented cases. It can be observed that Case 1 demonstrates robust model performance. However, when the noise is removed in Case 2 (where all data from Class 0 still originated from a single medical center), there is a notable decrease in validation accuracy. This suggests that the model in Case 1 may have been shortcutting, leveraging the

noise as a confounding factor rather than learning from the true underlying patterns. This is more evident when examining Case 3, where both noise and site-specific biases were removed, and we observe that the model is not able to generalize. This indicates that the site-specific characteristics, similar to noise, serve as a confounder, and impact the model's ability to generalize.

Table 4 further validates these observations. It illustrates the models' generalization performance on the external dataset (ex-test) compared to the test set. Note that the test sets for each case were developed under the same conditions as their corresponding training and validation sets, while the ex-test set was compiled without confounders and remained consistent across all cases. Notably, while Cases 1 and 2 show high accuracy on the test set, their performance on the external test set (ex-test) is significantly lower, confirming the challenges in generalization when noise and site-specific characteristics act as confounders. The observed difficulty in achieving consistent generalization across the validation, test, and external test sets in Case 3 suggests a fundamental challenge in the model's ability to learn from the data when confounders are controlled for.

The observed discrepancies in performance across the cases and the observed generalization gaps underscore the inherent challenges posed by black box models in ML as described in black box models pitfall (P10). To address this black box challenge and gain insights into the model's decision making process, we use saliency maps generated using the tf-keras-vis library (Kubota, 2023). Figure 6 provides these saliency maps for the models developed in each case of the experiment and visualizes the regions within the MRI slices the models considered important for making predictions. The map for Case 1 in Figure 6a shows a spread of highlighted regions across the background. It shows that the model is struggling to find correlations in the expected brain regions, and instead relies on spurious correlations in the noise present in the image as distinguishing features, and particularly so in the background. Since the brain regions already contains other inherent noise and since the background signal is otherwise zero, it is likely easier for the model to find spurious correlations in the background noise. For Case 2, where the images are from a single medical center but without added noise, the saliency map in Figure 6b is less scattered compared to Case 1 but still shows areas of focus outside the brain regions, which may be related to site-specific characteristics of the images. For Case 3, the saliency map in Figure 6c is expected to focus on the brain region without the influence of noise or site-specific biases. However, the map shows a diffused pattern of highlighted areas that may explain the model's poor generalization performance.

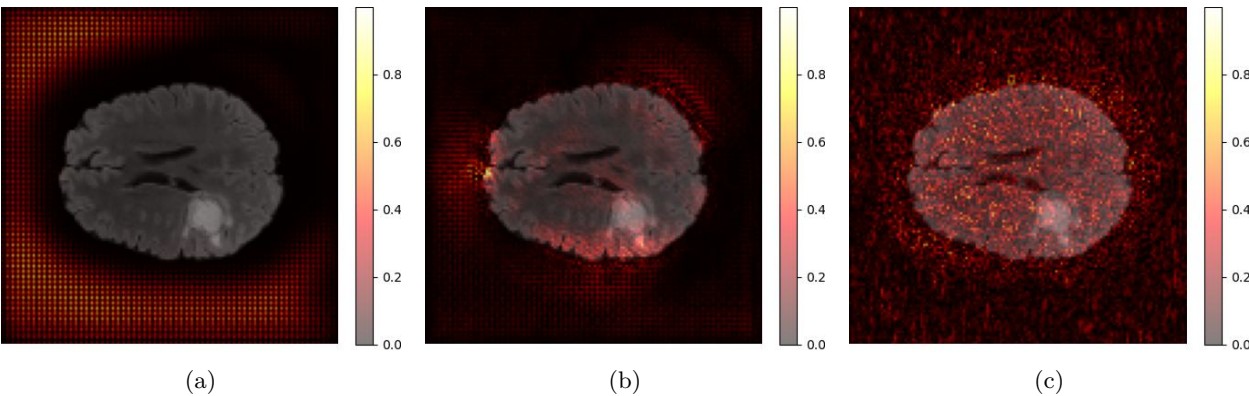

Figure 6: The obtained saliency maps for the three cases in Experiment 2. (a) Case 1: data with noise as confounder. (b) Case 2: data without noise but with site information as confounder. (c) Case 3: random data without confounders.

The saliency map for Case 3 in Figure 6c suggests that the model is struggling to identify robust features across the dataset that correlate with the MGMT methylation status. It could also be the case that the methylation status is a molecular feature, and while it may have some subtle manifestations in MRI scans, these might not be robust or consistent enough for a model to learn from. This latter point is supported by recent studies on ML models for MGMT methylation prediction (Kim et al., 2022; Saeed et al., 2023), where a significant majority failed to outperform chance levels in external validations. This point touches on

an important question in ML: Whether certain complex outcomes (like methylation status) can be reliably inferred from available data. This emphasizes the need for careful model selection and a deeper understanding of the data, particularly in tasks where the relationship between data and outcome is not straightforward.

### Experiment 3—Information Leakage

In this experiment, we focus on the information leakage pitfall, specifically examining the "data slicing" and "Pre-processing" aspects described in P6. We used the BraTS dataset to analyze how slicing 3D MRI scans into 2D images, without properly accounting for the correlations between these slices, leads to data leakage.

**Data and task:** In this experiment, we used the same dataset as in Case 2 of Experiment 1, derived from the BraTS 2023 Adult Glioma segmentation dataset. We selected 100 patients from the six sites previously mentioned. The same pre-processing steps as in Experiment 1 were applied to the extracted 2D slices. However, unlike in Experiment 1 where we extracted every third slice, here we extract every other slice from each patient, resulting in 50 slices per patient, and 5000 slices in total.

For the analysis, we adopted three different approaches to divide these slices into training, validation, and test sets. In the first approach, we allocated the slices on a patient-wise basis resulting in a dataset we refer to as the "Leakage-free data". Specifically, slices from 60 patients (3000 images) were used for training, while those from 20 different patients were allocated to each of the validation and test sets. This patient-wise division ensured that the training and test sets were independent, with no overlap in patient data, thus avoiding the risk of data leakage. The second approach, which introduced the risk of data leakage, involved shuffling all 5000 images and randomly allocating 3000 for training, 1000 for validation, and 1000 for testing. This method, which creates what we refer to as the "Single-leakage data", does not account for patient-specific correlations, potentially placing slices from the same patient in both training and test sets. The third approach is similar to the second, but with a variation in the pre-processing step. Instead of normalizing each slice individually using its maximum value for min-max normalization, we use the maximum value of the entire MRI volume of a patient to normalize all corresponding slices. This method introduces another potential source of data leakage, as described in the "Pre-processing" aspect of P6, and the produced data is referred to as the "Dual-leakage data".

Additionally, an external test set, termed "ex-test", was also included, that comprised slices from 20 different patients sourced from a site not included in the original six sites. This ex-test set served as an independent external test to evaluate the generalization gap, similar to the approach taken in the first two experiments.

**Model design:** In this experiment, we use the exact same model design as in Experiment 1, including the U-Net architecture, input configurations, optimizer, hyperparameters, and training methods. The same combined loss function and evaluation metrics were also used in the comparisons.

**Result and discussion:** Table 5 displays the segmentation performance for the three created datasets, evaluated using the DSC, HD95, RAVD, and ASSD metrics. For the Leakage-free data, the results show consistency between the test and ex-test sets with minimal variations in metrics, aligning with expectations from a dataset designed to prevent leakage. This consistency implies more reliable and generalizable model performance.

In contrast, the Single-leakage data demonstrates a higher DSC and lower HD95 in the test set compared to the Leakage-free data. This seemingly improved performance for this data is likely due to leakage from shuffling slices, which may cause the test data to be similar to the training data. However, the independent ex-test set shows a performance decline, emphasizing a reduction in the model's true generalization ability.

The Dual-leakage data reveals the highest DSC in the test set, indicating potential overfitting due to the additional pre-processing step involved in leakage. This overfitting is more apparent in the ex-test set results, where a significant drop in DSC and increases in HD95 and ASSD are observed, leading to a pronounced generalization gap.

Table 5: Obtained segmentation results for Experiment 3 (Leakage) measured by DSC, HD95, RAVD and, ASSD scores. The table shows the mean metric values, averaged over all images in each dataset, and the numbers in parentheses indicate the standard errors. Ex-test serves as an independent (external) test set, and the difference between test and ex-test results represents generalization gap.

|  |  | DSC | HD95 | RAVD | ASSD |
|---|---|---|---|---|---|
| Leakage-Free Data | Test | 0.81 ($\pm$0.01) | 1.60 ($\pm$0.16) | -0.03 ($\pm$0.07) | 0.20 ($\pm$0.06) |
|  | Ex-test | 0.82 ($\pm$0.01) | 2.16 ($\pm$0.19) | -0.01 ($\pm$0.05) | 0.44 ($\pm$0.10) |
| Single-Leakage Data | Test | 0.88 ($\pm$0.01) | 0.74 ($\pm$0.06) | -0.03 ($\pm$0.01) | 0.12 ($\pm$0.01) |
|  | Ex-test | 0.81 ($\pm$0.01) | 2.01 ($\pm$0.16) | 0.05 ($\pm$0.07) | 0.40 ($\pm$0.08) |
| Dual-Leakage Data | Test | 0.89 ($\pm$0.01) | 1.02 ($\pm$0.11) | -0.01 ($\pm$0.02) | 0.20 ($\pm$0.04) |
|  | Ex-test | 0.77 ($\pm$0.01) | 3.13 ($\pm$0.26) | 0.20 ($\pm$0.09) | 0.97 ($\pm$0.14) |

## 4 Review

In this section, we present the results of the review performed to determine the prevalence of unclear reporting regarding some of the ML pitfalls previously discussed, specifically within the context of segmentation tasks in the academic literature. We focused on two key conferences: the ICCV, known for its high-quality ML papers in computer vision, and the MICCAI, which specializes in medical applications of ML. These conferences allowed us to explore the issues across both general and specialized domains.

**Paper selection:** We reviewed the past 10 years of conference proceedings from MICCAI (2013–2022) and ICCV (2013–2021), with ICCV being biennial. From each conference edition, we initially extracted all papers with "segmentation" in the title. We then randomly selected 10 papers from each year's collection. Finally, we excluded papers that did not include ML research. More specifically, we excluded papers that did not involve a learning component, and for which the questions in the questionnaire were thus not applicable. This process resulted in a pool of 87 MICCAI papers and 39 ICCV papers, totaling 126 papers.

**Review process:** The review team consisted of five members, each with ML expertise ranging from 5 to 16 years. Each paper was evaluated independently by two different reviewers. To systematically assess the papers, we formulated a questionnaire comprising 14 targeted questions, each with multiple-choice options. These questions were designed to cover all the four different steps of the ML process, including three questions related to the data handling, seven questions focusing on the data leakage pitfall in the model design step, and the remaining four questions were devoted to pitfalls in evaluation and reporting. The questions were specifically designed to be answerable from the text of the papers, particularly the experimental sections, within a limited time frame.

To simplify the review process and facilitate objective assessment, we developed a detailed review protocol. This protocol included clear guidelines and examples for each question, ensuring consistency and minimizing subjectivity in responses. Additionally, to refine the questionnaire and eliminate any ambiguities, we conducted a preliminary test review with multiple team discussions. As part of the protocol, reviewers were instructed not to spend more than ten minutes per paper, focusing primarily on the experimental sections. This constraint intended to measure how easy it was to find the necessary information in each paper.

Each paper's review resulted in two sets of responses. We anticipated and indeed observed discrepancies between reviewers' answers. Rather than reconciling these differences, we chose to present the rate of discrepancy in our results. We believe that although the discrepancies in the reviews may partly stem from individual interpretation errors, yet they also likely reflect broader issues in the reporting standards within the ML community. The questionnaire file containing the title of all reviewed papers, the questions and answers along with a protocol file that describes the review details for each question are provided in the supplementary material. Note that, to have a precise review, each question originally had multiple detailed and granular answer options in the questionnaire. To ensure consistency and ease of interpretation, these original answer options were later condensed into four broader categories as explained in the following. The

original answers, along with the process of how they were mapped to the format presented in this paper, are detailed in the protocol file.

**Results and discussion:** Figure 7 presents the results of the performed review. The stacked bar plots on the left side of the figure depict the distribution of responses for each of the thirteen questions, with the vertical axis labeling the questions from Q1 through Q13. Each question had four potential responses: "Yes" (blue bars) indicates the paper adequately addressed the issue in question, reflecting a positive aspect; "No" (red bars) suggests the paper may have overlooked a critical issue, indicating a potential pitfall or a negative aspect; "Not Clear" (orange bars) denotes that the paper did not provide enough information to give a clear answer; and "Not Applicable" (gray bars) implies the question did not apply to the paper, such as when data augmentation was not part of the study. The right-hand plot shows the discrepancy rates between the two reviewers for each question. The lower bars in both plots correspond to results from MICCAI papers, while the upper bars are for ICCV papers. Note that for each paper we had two answers, and the answers were aggregated by assigning half a point to each reviewer's answer. When both reviewers agreed on a response, the answer received a full point towards the overall distribution. In cases of discrepancy, where reviewers did not agree, each divergent answer received half a point.

The first three questions (Q1–Q3) are about data handling and are related to P2. It is evident that most papers from ICCV and MICCAI (over 90%) clearly state their data set sizes. However, in this case even small rates of "No" responses are noteworthy, considering the importance and ease of reporting data size. "Not Clear" responses and discrepancies may stem from ambiguous details about the sizes of training, validation, and test sets, or post-processing data sizes. Regarding public data usage (Q2), ICCV papers tend to utilize public datasets more frequently compared to MICCAI, which aligns with the often private nature of data in medical studies. For Q3, focusing on dataset stratification, a majority of papers (over 80%) from both conferences did not explicitly mention stratifying their datasets. It is important to note that not all studies can feasibly implement stratification. In our review, we selected "Yes" even when authors considered and discussed the importance of stratification in their results and conclusions without practically implementing it.

Questions Q4–Q9, which assess potential data leakage are related to P6. As expected we can see a considerable number of "Not Applicable" responses. While there were not many direct "No" answers, the high rates of "Not Clear" responses and notable discrepancy rates suggest a lack of attention to leakage issues in reporting practices. This implies that the community may not fully recognize the importance of detailing how they prevent data leakage, which is critical for ensuring reliable results and reproducibility. The responses to Q10, evaluating the independence of the test set, were directly influenced by the answers to Q4–Q9; a "No" response to any of these questions resulted in a "No" for Q10.

Question Q11 is about fair hyperparameter tuning for competing methods and is related to P11. Over half the papers, especially from MICCAI, did not tune the hyperparameters of competing methods. High rates of "Unclear" responses and discrepancies highlights a similar issues as data leakage. That is, fair hyperparameter tuning is usually overlooked or poorly described.

Question Q12 focuses on reporting uncertainties in results and is related to P11 and P12. A significant proportion of papers, particularly from ICCV (over 90%), failed to account for uncertainties that can compromise the reliability of the results. For a deeper view of the results of Q12, we carried out an additional question to identify how the uncertainties were typically computed and reported. As shown in Figure 8, the scarcity of uncertainty reported in ICCV papers is apparent. MICCAI papers that report uncertainties tend to do so in the context of test set evaluations, with a smaller portion addressing them in cross-validation (or other resampling methods), independent test sets and "Other", which could for instance include a formal statistical test.

Finally, Q13 is about code sharing and is related to P14. We observe that only a modest portion of papers make their codes available, with this being more pronounced in MICCAI, limiting the reproducibility of the studies.

We now break down the obtained discrepancy rates into different categories: "Yes *vs.* No," "Yes *vs.* Not Clear," "No *vs.* Not Clear," and "Not Applicable (NA) *vs.* Others." Figure 9 presents the detailed breakdown

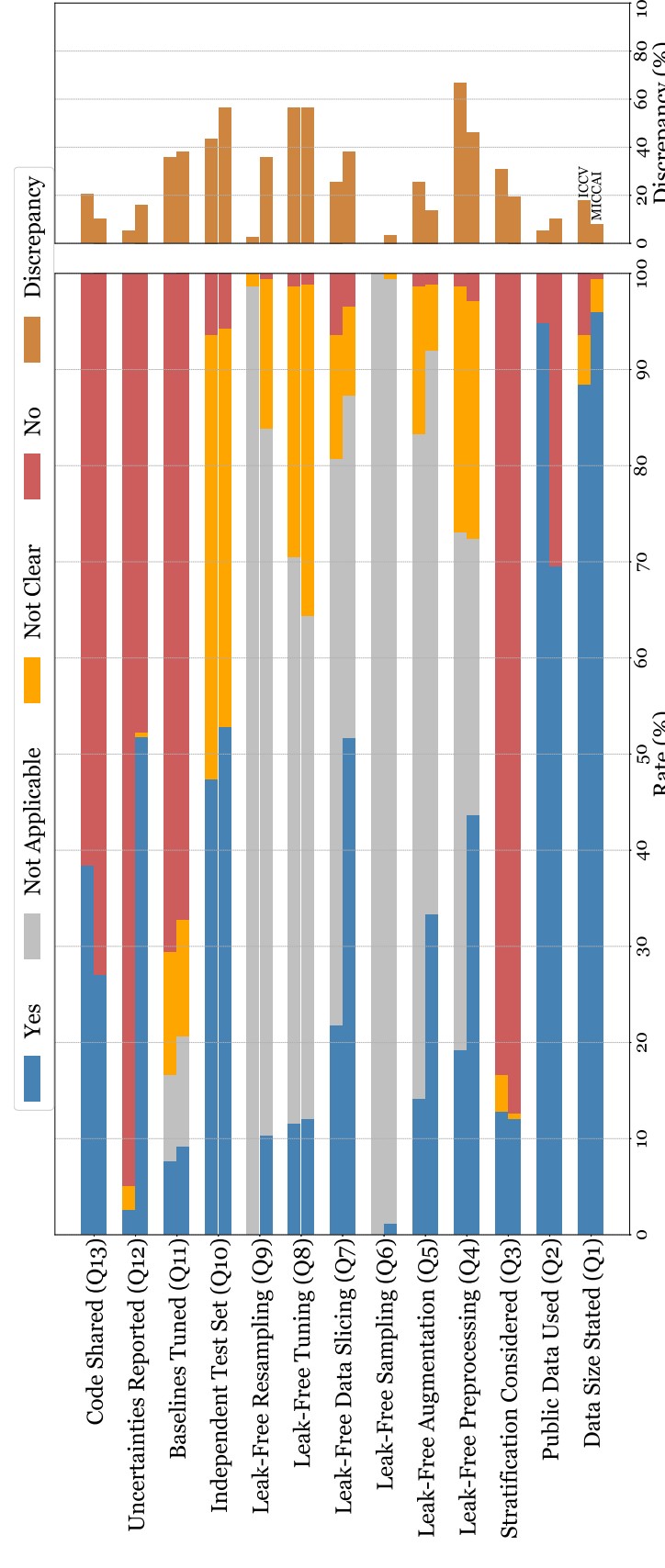

Figure 7: Review results. The below plot shows the response distribution for review questions (Q1–Q13). "Yes" (blue) indicates issues were adequately addressed; "No" (red) points to potential oversights or pitfalls; "Not Clear" (orange) reflects insufficient information for a definitive answer; and "Not Applicable" (gray) denotes irrelevance to the study. The upper plot illustrates the discrepancy rates between reviewers' answers. Lower bars relate to MICCAI papers, while upper bars are for ICCV papers.

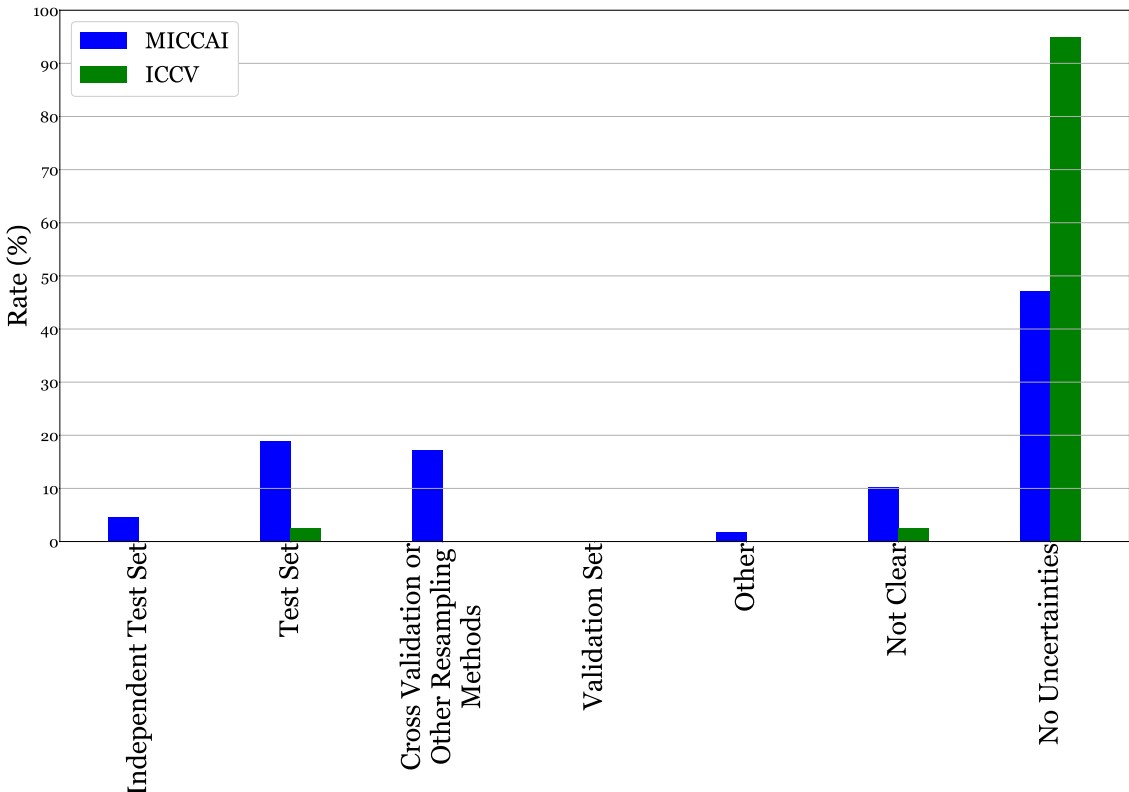

Figure 8: The obtained review results for the additional question of "How and on what subset of data the uncertainties are computed?".

of these discrepancies for both the ICCV and the MICCAI papers. According to the figure, for Q1, Q2, Q3, Q12, and Q13, the rate of "Yes *vs.* No" discrepancies is the highest for both ICCV and MICCAI (except for Q1 in MICCAI). Part of these discrepancies can be attributed to the limited time available for review, but they can also be linked to the absence of adequate information, leading to varying interpretations. For example, in Q1, which asks whether the dataset size is stated, there might have been some mention of data size, but not in a precise or clear manner. For instance, the number of videos used in a study might be mentioned, but the actual number of image frames, which is what the model was training on, could be left ambiguous for the reader to infer. This lack of clarity could result in different interpretations between reviewers, contributing to the observed discrepancies.

For Q11, which asks whether the hyperparameters of competing methods were tuned, the "No *vs.* Not Clear" and "NA *vs.* Others" discrepancy rates are higher compared to the other types of discrepancies. These discrepancies likely stem from the fact that in many cases, there was no explicit mention of the hyperparameters for competing methods. This omission left reviewers uncertain as to whether there were no hyperparameters to tune (making the question not applicable), or if there were hyperparameters, but they were simply not tuned.

For Q4–Q9, which focus on questions related to data leakage, the "NA *vs.* Others" discrepancy rate is higher in both ICCV and MICCAI papers. Beyond the time constraints of the review process, these discrepancies can largely be attributed to the fact that most papers provided little to no information on leakage-related processes, such as preprocessing, slicing, or other critical steps. This lack of clarity made it difficult for the reviewers to determine whether these processes were applied. In some cases, a reviewer might infer, based on the nature of the problem, that certain steps like preprocessing must have been conducted, but without explicit mention in the text, this remained unclear, leading to discrepancies in interpretation.

**Practical insights from review findings:** Throughout this paper, we have offered brief suggestions on approaches to address each identified pitfall, referencing studies that discuss potential solutions. However, our primary aim was to raise awareness by highlighting these pitfalls rather than providing concrete methodologies. It is also important to recognize that, although each pitfall is presented independently here, in practice, they are often deeply interconnected. For instance, issues like dataset shift, spurious correlations, and overfitting can reinforce one another, as can overfitting and information leakage. Developing practical, comprehensive solutions for each pitfall or offering specific steps requires further, in-depth investigation beyond the scope of this study.

However, drawing on insights from our review of the ICCV and MICCAI papers, we would like to offer some practical suggestions that could serve as practical first steps for researchers. For instance, we observed that stratification and data control practices are frequently neglected. This could indicate a lack of attention to understanding and managing shifts and biases within datasets. Addressing this oversight by carefully examining data and actively controlling for shifts and biases may help mitigate issues like spurious correlations and overfitting. Additionally, a lack of clarity around data leakage prevention suggests that leakage is often overlooked, a significant problem given its impact on model reliability. To improve practice, researchers should be meticulous, transparent, and thorough in implementing and documenting leakage-related processes.

Another key area is hyperparameter tuning, where we noticed considerable ambiguity. This lack of attention to optimizing hyperparameters can lead to overfitting or underfitting, potentially resulting in unfair comparisons with competing models. A robust approach to hyperparameter tuning is essential for fair and accurate model evaluation. Finally, we found that uncertainty quantification is frequently ignored. Including uncertainty estimates could be a valuable, actionable step toward improving model reliability. Finally, prioritizing clarity, transparency, and detailed reporting can have a substantial impact on mitigating these pitfalls and enhancing the generalization of ML models.

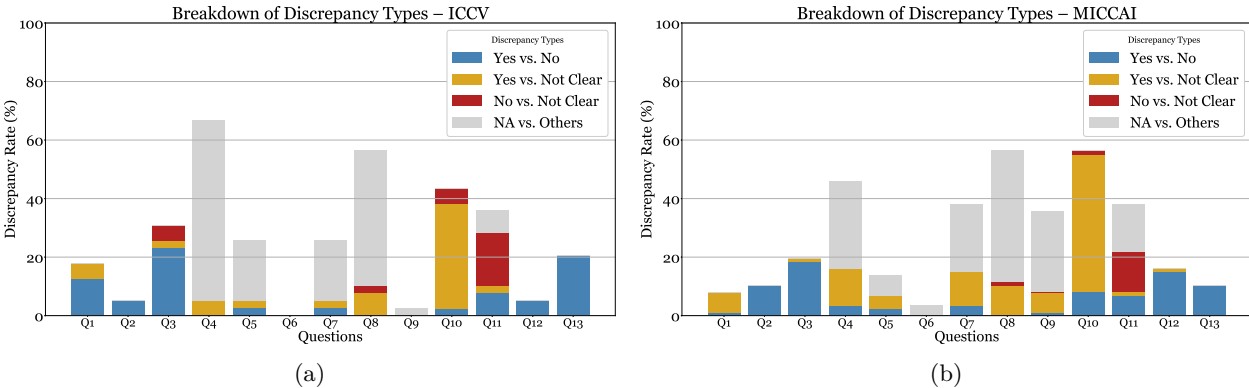

(a)                                                                 (b)

Figure 9: Breakdown of discrepancy types for ICCV and MICCAI papers. Discrepancies are categorized as "Yes *vs.* No," "Yes *vs.* Not Clear," "No *vs.* Not Clear," and "Not Applicable (NA) *vs.* Others."

## 5 Conclusion

This paper includes three parts. In the first part, presented in Section 2, we discussed 14 common technical pitfalls compromising the generalizability of ML models. We presented these pitfalls by associating them to the four main steps of the ML pipeline.

Utilizing the BraTS dataset, we performed three experiments to show the impacts of these pitfalls on a segmentation task. These experiments were focused on Dataset Shift (P2), confounders (P3), and Information Leakage (P6), while also touching upon other issues such as Misused Metrics (P9) and the challenges of Black Box Models (P10) wherever applicable. The results from these experiments clearly demonstrated how these pitfalls, if not adequately addressed, could skew the results and affect the model's generalization capabilities. It is important to note that the conducted experiments does not encapsulate the entire range of pitfalls, tasks, and datasets in ML. The use of a single dataset and a single model architecture, UNet, further limits

the interpretability of the results. Exploring additional pitfalls, other ML tasks, and diverse datasets in future studies could provide a better understanding of the challenges in ML research.

Furthermore, our study included a review of 126 papers randomly selected from the ICCV and MICCAI conferences over the past decade. This review, conducted using a structured questionnaire answered by five reviewers, aimed to assess how clearly recent ML research addresses these pitfalls. The review revealed a notable oversight of the pitfalls within the ML community. Particularly, questions related to information leakage were not clearly answerable in the papers, suggesting a widespread tendency within the community to overlook these important issues. Note that this review, containing 150 papers from ICCV and MICCAI, represents only a fraction of the ML research. Despite randomize selection, this sample may not fully reflect the broader issues and practices in the ML field. Moreover, the review results rely on the interpretations of reviewers responding to the questionnaire.

In conclusion, we emphasize on the necessity for collective efforts within the ML community to identify and mitigate these pitfalls. Designing rigorous benchmark data, evaluation metrics and procedures, and transparent reporting standards can improve research integrity and reproducibility. In addition, the community can incentivize researchers to address these pitfalls by using detailed review processes that prioritize methodological rigor and clear reporting. Promoting best practices and robust research design principles through enhanced training and peer-review guidelines can also improve accountability and precision in ML research.

### Acknowledgments

We are grateful for the financial support obtained from The Swedish Childhood Cancer Fund (Barncancerfonden, MT2021-0012) and Lion's Cancer Research Foundation in Northern Sweden (LP 22-2319 and LP 24-2367).

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
