# OpenReview forum: "From Promise to Practice: A Study of Common Pitfalls Behind the Generalization Gap in Machine Learning"
_TMLR — Accepted by TMLR_

### Review · Reviewer_Jn3Q · 2024-09-16

**Summary Of Contributions:**

The authors summarize into 14 categories relevant pitfalls in current machine learning (ML) practices: spanning from dataset creation to interpretation and reproducibility. These pitfalls were identified through a literature review of 60 papers. Following their presentation, 3 of these pitfalls are demonstrated through experiments on a brain tumor segmentation dataset. Finally, a second experiment, encompassing 126 papers from two conferences, ICCV and MICCAI, is conducted to quantitatively evaluate the quality of reporting (and some other pitfalls) in current scientific papers.

**Audience:**

Yes

**Broader Impact Concerns:**

Concerns are addressed sufficiently, and experimental limitations are clearly stated.

**Claims And Evidence:**

Yes

**Requested Changes:**

## Main
- I believe more than citing work about methods on how to deal with (some of) the pitfalls, this paper could greatly benefit to go beyond merely the citation and add a small analysis. One of the examples is ending P11 with "general recommendations and best practices were reviewed by Demsar (2006)". This can be discussed further and provide a more recent insight to the given recommendations.
- More details and discussion of the results should be added for the methodology in the Review section. For instance, in the questions asked, Q4 is about 'Leak-free preprocessing', and the possible responses are "No preprocessing/Preprocessing separately after splitting/Preprocessing the entire data before split with no risk of leakage/Preprocessing the entire data before split with risk of leakage/Not clear". While the first and last options are straightforward to categorize as "Yes/No", it's unclear whether the other responses are all considered as "Yes" or "Not clear". This issue also applies to several other questions. Results could also be discussed more. For example, is the discrepancy more pronounced between Yes and No, or between Yes and Not Clear, or is it uniform? These distinctions have different implications—one may indicate a timing issue in the identification (given the 10 minutes per paper), while the other may point to an interpretation issue.
- Add reproducibility parameters for experiments. For example, in experiment 1, what was the learning rate (was it tuned? It seems that in experiment 2, it was not)? Regarding early stopping, it was mentioned that it was used when performance "didn't improve for 10 epochs," but what specific criteria were applied (threshold value, relative or absolute evolution)? Additionally, the loss function involves a balance between cross-entropy and Dice loss with weights of 0.3 and 1-0.3=0.7. How was it determined or optimized? In the cited Vu et al. (2021), their weights (which differ from 0.3 on another dataset) were established after a hyperparameter search. Finally, what dropout rate was used for the Dropout Layer in experiment 2? These information should be added.

## Not critical

- Both the "labeling mistakes" and "data augmentation" parts could benefit referencing at least a paper like other paragraphs of the same type for readers. E.g. [1], [2] and [3] (or others)
- In the "labeling mistakes" part, the paper acknowledges that humans can mislabel data, but ignores the impact of the aggregation strategy on the labeling mistakes. When multiple humans/machine label an image, disagreements can be discarded or modeled. Taking the majority vote or model confusions using *e.g.* Dawid and Skene's aggregation [5] can lead to different labels and result in different generalization gap.
- In the overfitting/regularization/double descent part, it could be of interest to add that the double descent can be mitigated with a regularization in some cases [4] as this would link back to the paragraph above.
- To better connect section 1 and section 4 could the group of questions used for the review clearly be linked to the original pitfalls presented in section 1 with the PXXX notation presented?

## Typos and duplicated text:
- Page 16: the paragraph starting with "Furthermore" ending with "applicability" is repeated twice.
- Page 19: same remark with the sentence "the number of epochs..."
- Page 7: Oakden-Rayner et al. (2020) after 'splitting the data' should be a reference in parenthesis
- Page 20: paragraph 2, "case 1" should be capitalized following the introduced notation

## References

- [1]: Goh, Hui Wen et al. (2022) (*Utilizing supervised models to infer consensus labels and their quality with multiple annotators*)
- [2]: Northcutt, Curtis G., Anish Athalye, and Jonas Mueller. (2021) (*Pervasive label errors in test sets destabilize machine learning benchmarks.*)
- [3]: Xu, Yan, et al. (2016) (*Improved relation classification by deep recurrent neural networks with data augmentation*)
- [4]: Nakkiran, Preetum, et al. (2020) (*Optimal regularization can mitigate double descent*).
- [5]: Dawid, A.P. and Skene, A.M. (1979), (*Maximum Likelihood Estimation of Observer Error-Rates Using the EM Algorithm*). Journal of the Royal Statistical Society

**Strengths And Weaknesses:**

## Strengths

- The paper details each pitfall and contextualizes them. They are clearly identified and visually grouped in Fig.1. We clearly understand the main contribution of the paper
- The paper is well organized with an extensive literature review, and flows nicely, especially in the main contribution: Section 2
- Experiments show examples of how important it is to understand the selected pitfalls and avoid them. Each pitfall selected for the experiments is isolated.

## Weaknesses

- The paper classifies the pitfalls but lacks methodologies to handle them (and acknowledges it). While the general insights are of interest as they clearly highlight currently open problems, adding more analysis of the few presented tools to avoid them could be of high value.
- The methodology used in the Review section lacks clarity regarding how the reviewers' answers were converted into the Yes/Not Applicable/Not Clear/No format for some questions. Additionally, the discussion over the results could be expanded.
- This paper addresses reproducibility issues in ML research; however, the experiments section lacks sufficient details for replication.

---

> ### Author Response · Authors · 2024-10-08
> **Response to Reviewer Jn3Q**
>
> We thank the reviewer for their review of our manuscript. We appreciate the recognition of the strengths of our work. We also agree with the reviewer’s comments regarding the areas that needed improvement, particularly related to the reproducibility and the clarity of the review process details. In the following, we respond each of the concerns raised.
> ## Requested changes: main
> **Comment1: Analysis for P11**
>
> We acknowledge the reviewer's concern and in response, we have expanded our discussion from Demsar(2006) on statistical tests used to evaluate model performance. We now have added a discussion in P11 of methods, including the Wilcoxon signed-rank test, paired t-test, Friedman test, and ANOVA, emphasizing their appropriate contexts and applications.
>
> **Comment2: Clarify categorization of review responses and discrepancy types**
>
> 1. We thank the reviewer for highlighting this important point that we had missed to include. We have now added this information to the *protocol file*. For each original possible response, the corresponding category of the final answer is provided. For example, for Q4, "Preprocessing the entire data before split with risk of leakage" is categorized as "No", because based only on the text of the paper we see a risk for a potential pitfall. We also added a clarification regarding this in the paper.
>
> 2. To expand the discussion, we have broken down the discrepancy rates into four categories: "Yes vs. No," "Yes vs. Not Clear," "No vs. Not Clear," and "Not Applicable vs. Others." Figure 9 has been added to display this breakdown for ICCV and MICCAI papers. For Q1, Q2, Q3, Q12, and Q13, "Yes vs. No" discrepancies were highest, likely due to timing issues or ambiguous information. For example, for dataset size, there may be some mention of data size, but not in a precise manner. For instance, the number of videos may be mentioned, but the actual number of image frames, which is what the model was training on, could be left ambiguous for the reader to infer. Questions Q4–Q9, related to data leakage, showed increased "NA vs. Others" discrepancies, often because most papers provided little to no information on leakage-related processes like preprocessing or slicing. This lack of clarity made it difficult for reviewers to determine whether these processes were applied. In some cases, a reviewer might infer, based on the problem, that certain steps like preprocessing were conducted, but without explicit mention, this remained unclear, leading to discrepancies in interpretation. These points are discussed in more detail in the revised manuscript.
>
> **Comment3: Reproducibility parameters for experiments**
>
> We added the following clarifications to the paper:
> 1. The learning rate was set to $5\cdot10^{-5}$ after a grid search over $\{5\cdot10^{-3}, 1\cdot10^{-3}, 5\cdot10^{-4}, 1\cdot10^{-4}, 5\cdot10^{-5}, 1\cdot10^{-5}, 5\cdot10^{-6}, 1\cdot10^{-6}, 5\cdot10^{-7}, 1\cdot10^{-7}\}$, which yielded the best validation accuracy.
>
> 2. The models were trained for up to 100 epochs, with early stopping applied when the validation loss failed to improve for 10 consecutive epochs. Improvement was determined by a decrease in the validation loss relative to the best value observed so far.
>
> 3. The binary cross entropy was weighted at 0.3 and Dice loss at 0.7, with the 0.3 value for binary cross entropy chosen based on a grid search over the values $\{0, 0.1, 0.3, 0.5, 0.7, 0.9, 1\}$.
>
> 4. The dropout rate was set to $0.5$ after a search over values $\{0.4, 0.5, 0.6, 0.7\}$
>
> ## Requested changes: Not critical
> **Comment1: Labeling mistakes and augmentation**
>
> We found the mentioned papers very helpful and have included them for both the "labeling mistakes" and "data augmentation" sections, as suggested.
>
> **Comment2: Aggregation strategy on labeling mistakes**
>
> We have expanded the "Labeling mistakes" section to include a discussion on the importance of aggregation strategies when multiple annotators label the same data. We now discuss aggregation methods, such as majority voting and more sophisticated approaches like Dawid and Skene's strategy.
>
> **Comment3: Regularization and double descent**
>
> A discussion on how regularization can mitigate the double descent phenomenon has been added, referencing Nakkiran et al. (2020).
>
> **Comment4: Link review questions to original pitfalls**
>
> We now added the related pitfalls for each question when discussing the results of each question. Specifically, questions Q1-Q3 are related to P2-Dataset shift, questions Q4-Q10 are related to P6-Information Leakage, question Q11 is related to P11-Baseline Comparisons Issues, question Q12 is related to both P11-Baseline Comparisons Issues and P12-Insufficient Reports, and finally question Q13 is related to P14-Code Reproducibility Issues. These relations have now been made clear in the manuscript.
> ## Typos and duplicated text
> We thank the reviewer for noting the typos and duplicated text. These issues have been corrected.

---

> > ### Comment · Action_Editor_gY2X · 2024-10-15
> >
> > Dear authors,
> > If you modified the pdf, can you upload it directly, so that the reviewer can check the modifications? I suggest including them in a different color.
> > Thanks,
> > Your AE

---

> > > ### Author Response · Authors · 2024-10-22
> > >
> > > The revised manuscript is uploaded now and the changes are highlighted based on the reviews of both Reviewer Jn3Q and Reviewer ncKj.

---

> > ### Comment · Reviewer_Jn3Q · 2024-10-23
> >
> > Thank you for updating the paper, protocol file and for answering my questions. The added discussion in P2 is also of great interest.
> >
> > Just a small spacing issue at the beginning and end of the sentence added in the Model design paragraph left.

---

> > > ### Author Response · Authors · 2024-10-23
> > >
> > > We appreciate your feedback, which we believe led to improvement in our paper. Thank you for catching the spacing issue. We have now fixed it in the new uploaded version.

---

### Review · Reviewer_ncKj · 2024-10-11

**Summary Of Contributions:**

This paper discusses various failure modes in ML processes, with a special emphasis on medical imaging. They begin with a categorization of the ML pipeline into 4 stages, each with their own failure modes - 14 in total. These range from effects like measurement error in the data collection process, to code reproducibility issues in reports. The authors describe these modes in detail, and then investigate a few of them using the BraTS dataset, with a series of experiments that reflect different types of process failures that can affect ML pipelines.

The authors conclude with a survey of papers in ICCV and MICCAI, and analyze the presence of all 14 failure modes. They concluded that many failure modes are prevalent, especially a lack of discussion around data leakage, poor hyperparameter tuning of baselines and competing methods, and failure to account for uncertainty. The authors conclude with a call to action for the community to improve standards around all aspects of the data pipeline.

**Audience:**

Yes

**Claims And Evidence:**

Yes

**Requested Changes:**

Add a discussion of papers [1,2,3] listed in "strengths and weaknesses".

There are some typographical errors and language issues in the text; e.g. "the number of epochs was set to 100 as a maximum limit, with early stopping employed" begins on its own paragraph. Use the `\citet{}` command for in-sentence referencing (e.g. "can be found in (Barocas et al., 2017; Bender et al., 2021; Schwartz et al., 2022)." (or, rewrite the text so `\citep{}` is appropriate). Sometimes there is language redundancy around citations, please clean that up.

The authors mention the power of large models trained on large, multimodal datasets with regards to their generalization ability; however the authors also state:

"The current trend in ML, driven by the availability of large amounts of data and sophisticated methods, often encourages the belief that larger datasets and more computational power are essential for success... This belief can lead to using unnecessarily complex models when simpler ones might suffice"

I don't think this point is related to datasets. In general, having more data (without sacrificing data quality) seems to be generally beneficial. Indeed, the trend towards larger models started in part because larger models can often USE larger datasets more efficiently - as shown by the empirical phenomenon of neural scaling laws. It would be helpful for the authors to talk through this point more carefully.

In Figure 2, please indicate for each metric whether maximization or minimizaton is the goal.

How are the gap distributions computed in Figure 3?

Can the authors provide more explanation for what the experiments in Figures 2 and 3 tell us, given that 3 of the 4 gaps are insignificant?

**Strengths And Weaknesses:**

Overall, I believe this paper provides a nice survey and description of many common failure modes. The experiments are well designed and informative, with some caveats discussed below. The survey on works from the two conferences is also interesting and seems reasonably designed.

There is some question of the novelty of some of the observations of failure modes; however, having such modes in one place, combined with a survey in the segmentation literature makes the description of the failure modes interesting for the readers of TMLR.

One line of work that is not discussed in the paper but seems highly relevant are studies on overfitting in popular image datasets, where they construct new test sets [1,2,3]. These test sets are often found to cause accuracy drops, but the relative ranking of algorithms on original test sets remain consistent, suggesting that even with many potential issues of data leakage, overfitting, and improper test set construction, these benchmark datasets remain useful in designing more general models.

The authors also talk about i.i.d. datasets, and how deviations from i.i.d. can cause issues. I think it's worth pointing out that it's often not really clear what i.i.d. means in practical settings. Most if not all data are in some sense extremely undersampled from their "natural" distribution (as well as one can define it). Any notion of "biased sampling" or "off-distribution data" really need to be grounded in specific tasks or goals. The authors provide very clear and motivating examples in their studies on BraTS; having a more motivated and grounded discussion in earlier sections will help the main text quite a bit.

In the introduction, the authors mention that many large "foundation models" seem to bypass many generalization issues directly. However, the authors also caution against relying on large datasets. I've commented further on this in "Requested changes".

The experiments presented in Figures 2 and 3 are well motivated, but the results seem weak. Only one out of the 4 scores displays the trend hypothesized by the authors. Does the overall significance of the result hold up even when using e.g. a Bonferroni correction to account for the fact that 4 different measures were tested? In general the ex-test seems highly variable for most of the measures. Even if more samples were taken so that the p-value of the difference was reduced, the ratio of the difference to the variance remains low - which doesn't lend much support to the hypotheses of the paper.

One overall point which is underdiscussed is the difference between many academic works on "standard" datasets, and works which introduce new datasets. Indeed, in clinical and industrial settings the majority of the work involves taking data, assembling them into datasets, and even updating them in real time. This is quite different from how many academic works operate. More discussion of this point and how it affects the research landscape would be quite beneficial.

[1] https://proceedings.mlr.press/v97/recht19a.html
[2] https://arxiv.org/pdf/1806.00451
[3] https://proceedings.neurips.cc/paper/2019/hash/ee39e503b6bedf0c98c388b7e8589aca-Abstract.html

---

> ### Author Response · Authors · 2024-10-22
>
> We thank you for the valuable feedback. Below, we address each of the concerns raised by the reviewer and have made the corresponding revisions to the manuscript. The revised manuscript is uploaded now and the changes are highlighted.
> ## Strengths And Weaknesses:
> Here we only consider comments that are not repeated in the "Requested Changes" section.
>
> **Comment1: Discussion on IID in practical setting**
>
> This is an important point, and We have now added a paragraph in P2-Dataset Shift section to cover the point about the complexity and limitations of IID assumption in practical settings.
>
> **Comment2: Discussion on difference between many academic works on "standard datasets", and works which introduce new datasets.**
>
> We have added a paragraph to P1-Design-Lacking Data Collection section to address the difference between academic work on standard datasets and the data collection practices in clinical and industrial settings. We discuss the challenges of assembling and updating datasets in real time and how these factors impact model performance and generalization in practical applications.
>
> ## Requested Changes:
>
> **Comment1: Discussion on studies on overfitting in popular image datasets such as ImageNet and CIFAR-10**
>
> We agree on the relevance of this point and have added a discussion in P2-Dataset shift section where we reference these studies. We highlight how newly created test sets for ImageNet and CIFAR-10 reveal performance drops but maintain the relative ranking of models, supporting the continued importance of benchmark datasets despite concerns about overfitting.
>
> **Comment2: typographical errors and language issues**
>
> Thank you for pointing the typographical errors. We have gone through the manuscript to fix these issues.
>
> **Comment3: Clarification needed on dataset size and model complexity**
>
> We agree with the reviewer that larger datasets, if of high quality, are beneficial and can improve model performance. Our original phrasing in section P7-Model-Problem Mismatch implied a criticism of large datasets. To revise the text, we have removed the reference to large datasets and focus only on the potential issue of unnecessary model complexity.
>
> **Comment4: Clarification of metric goals in Figure 2**
>
> For the DSC metric, higher values indicate better performance, while for HD95, RAVD, and ASSD, lower values are preferable. This information has been added to the caption of Figure 2 and to the y-labels in the figures, using an upward arrow for the DSC score and downward arrows for the other metrics to indicate this.
>
> **Comment5: How are the gap distributions computed in Figure 3?**
>
> Thank you for pointing this out. The distribution of the differences in gaps between Case1 and Case2 is estimated using Kernel Density Estimation (KDE). This information has now been added to both the paper and the figure caption for clarity.
>
> **Comment6: Statistical significance of the results in experiment 1**
>
> We thank the reviewer for mentioning this point. In response, we decided to expand the experiment by increasing the number of datasets from 10 to 25 to provide a more clear and robust analysis. With this larger sample size, the results are now more easily discernible.
>
> As detailed in the revised text, the p-values for three metrics, i.e.,  DSC, HD95, and ASSD, are now safely below the chosen 0.05 significance threshold ($p = 1.31 \times 10^{-7}$, $ p = 5.94 \times 10^{-4}$, and $p = 3.60 \times 10^{-3}$, respectively), indicating statistically significant differences in the generalization gaps between Case1 and Case2. However, for the RAVD score, the p-value ($p = 8.79 \times 10^{-2}$) remains above 0.05, suggesting that this metric does not show a statistically significant difference between the two cases. To account for the fact that four different metrics were used, we Bonferroni corrected the significance threshold to $0.05/4=0.0125$. With this stricter threshold, the results for DSC, HD95, and ASSD remain statistically significant.
>
> The different behavior of the RAVD metric is discussed in the existing paragraph of the paper in Experiment 1 (highlighted now), where we explain that while DSC, HD95, and ASSD capture overlap accuracy and boundary delineation, the RAVD score reflects the model's ability to estimate tumor volume. Given that RAVD can show a perfect score even in cases of imperfect segmentation, we highlight the importance of using multiple metrics to evaluate different aspects of a model.
>
> Figure 2, Figure 3, Table 2, and Table 3 are changed accordingly.

---

### Review · Reviewer_UGJ4 · 2024-10-27

**Summary Of Contributions:**

The authors study common pitfalls that prevent machine learning systems from generalizing well and performing satisfactorily once in use on real data.

The paper is organized in 3 parts.

In the first part, the authors review 60 publications on machine learning pitfalls and compile a summary list of 14 pitfalls across 4 stages in the development of a machine learning pipeline. They provide a brief summary, references, and high-level recommendations for each pitfall.

In the second part, the authors focus on 3 of the identified pitfalls (dataset shift, confounders, and data leakage). They perform experiments on the Brain Tumor Segmentation dataset to demonstrate the result of incorrectly handling these 3 issues: optimistic evaluation scores and lack of generalization to truly unseen data.

In the third part, the authors review 126 papers from 2 machine learning conferences (ICCV and MICCAI) to study how the identified pitfalls are handled in the literature. 2 expert annotators answer a questionnaire about each of the 126 papers, providing information such as whether code was shared or whether an independent test set was used. The authors then comment on the results and the inter-annotator disagreements.

**Audience:**

Yes

**Claims And Evidence:**

Yes

**Requested Changes:**

The paper extensively discusses "confounders". This is a causal notion which does not directly apply to a predictive setting. I completely agree with the authors that understanding the causal effects which govern the system under study is crucial. This helps formulate hypotheses about likely distribution shifts between training data and data that will be seen after deployment.

However the existence of a confounder can only be defined in the context of estimating a causal effect. It does not directly translate to "spurious correlations" (a term that is used but could be defined more precisely in the paper). Moreover correlations that do not reflect a direct causal link can still be very real, useful for prediction, and generalizable. This example is used in the paper:

>    "suppose we want to develop a ML model to predict whether a person will buy a particular product based on income as input variable. In this scenario, education level can affect both the input and output variables and confound their relation"

This would be an issue if we were trying to estimate a causal effect such as "does income influence interest in this product", i.e. "would raising someone's income increase the chance that they will buy the product". It is also a relevant observation to reason about dataset shifts, for example if we have reason to believe that the influence of education on income might be different in the target population than in the training data. It is _not_, however, something that makes the correlation between the 2 "spurious", or that poses a problem for generalization if the training data is representative of the target population.

The interplay between causal relationships and generalization is therefore more complex than suggested by P3. See for example [Schölkopf, 2012, "On causal and anticausal learning"] for a discussion of this topic in a restricted setting. We cannot say that we need to "control for confounders", which is only defined when we are estimating a causal effect. Removing associations with a variable identified as a "confounder" in a predictive setting can hinder rather than help prediction and generalization. The authors recommend mitigating the effect of confounders and "producing residualized variables" and point to Jager et al. 2008. However this can be detrimental; moreover the reference focusses on estimating the (causal) effect of an exposure on an outcome ⸺not on prediction⸺ so a discussion of how the concepts discussed in Jager & al. can be adapted to a predictive setting is necessary.

Similarly, I find the discussion of Experiment 2, "Confounders", a little confusing. The causal graph and the variables that are considered as "confounders" are not made explicit until the "results and discussion" paragraph. There, it is stated that the "site-specific characteristics, similar to noise, serve as a confounder". In P3 a confounder has been defined as a variable that influences both the input and the output. Does this mean that the added Gaussian noise influences the output, i.e. adding noise to an image causes methylation of tumors?
Rather than confounding, what this experiment illustrates is sample selection bias, because Class 2 samples have been selected exclusively from Site 1 rather than sampled uniformly from the population of tumor patients.
Note that this does not diminish the value of Experiment 2, which is interesting and illustrates the risk of a model relying mostly on site effects and other associations created by preferential sample selection. This is indeed a real and frequent threat to generalization. My issue is only with the discussion around confounders which I believe should be clarified.

Section P1 also mentions experimental design and randomized controlled trials, which are usually discussed in the context of estimating the influence of a treatment on an outcome. There also, some more explanations on how these notions translate to machine learning would be useful.

An explicit discussion of the different levels of grouping (eg in the experiments, slice, scan, patient, hospital, region/country) across which a model is expected to generalize could be helpful.
When designing a learning system we need to decide what kind of generalization will be needed and the validation experiments should reflect that: we want to hold-out hospitals if we want to generalize to a new hospital but that may not always be the case.

## Minor comments

- Having a class imbalance is not a dataset shift if the prior probability of each class is the same in the training and target populations. Moreover if the model is well calibrated this may not require any special handling.
- Could the authors state why they chose those 3 pitfalls among the 14 for the experiments?
- Why was a very small proportion of the BRATS dataset used (dozens of patients in the experiments, over a thousand in the dataset)?
- The paper says the dataset contains 5,880 MRI slices; I believe it is 5,880 full brain scans rather than slices.
- In experiment 1, 25 datasets are sampled with different random seeds. Do the sampled datasets overlap? If so the results are not independent which could be an issue for the statistical tests described later.
- In experiment 3 "dual-leakage data": normalizing across the whole batch rather than slice-by-slice could change the results regardless of leakage. Therefore it might be more interesting to compare a case where normalization is done across the whole dataset vs across only the training data.
- About the same experiment: using the testing data for scaling in preprocessing is indeed an instance of data leakage. However using it for feature selection (or any other preprocessing operation that involves the target variable y) tends to result in much more dramatic over-estimation of test prediction scores. That could be an interesting pitfall to illustrate rather than normalization (which relates to my comment about prioritizing the pitfalls).
- Some of the pitfalls and review questions do not relate directly to generalization. For example sharing code is a somewhat different issue; a private model whose code and weights are not shared can generalize and perform well once deployed. It could be interesting if the authors could mention why they chose to include those aspects as well in the study.
- [Bernardi & al 2019. "150 successful machine learning models: 6 lessons learned at booking.com"] also discusses several of the pitfalls identified in this paper, but more from the perspective of recommendations and thus is quite complementary to this work so might be worth mentioning in my opinion.

**Strengths And Weaknesses:**

# Strengths

The paper provides a comprehensive, high-level overview of pitfalls that threaten the generalization of machine learning models. It is well documented with many references. Therefore it can serve as a reference or checklist when planning a machine learning project or auditing or improving an existing one. It can also be useful as a starting point on this vast topic, by categorizing the potential issues and providing references to learn more about any of them.

The paper also presents experiments on real data that illustrate the effect of several pitfalls. This is useful to convince that those are not only theoretical issues but practical concerns, and to illustrate how they may arise in a realistic setting.

The review of MICCAI and ICCV papers is interesting and shows that those issues are still very relevant in the recent literature. It helps identify particular topics that the machine learning community, authors and reviewers should pay more attention to by showing shortcomings of many publications in 2 well-regarded conferences -- for example the fact that most publications do not tune the hyperparameters of baselines, resulting in unfair comparisions with existing methods.

Overall the paper is well organized and easy to follow.

# Weaknesses

One potential drawback of tackling such a vast topic is that each pitfall has to be covered at a very high level. Some sections such as P1 may perhaps benefit from more concrete examples or some more technical points. However this is mitigated by providing many useful references for readers wanting to learn more.

The paper offers little guidance to ML practitioners wanting to improve the generalization of their pipelines. While it explicitly states that proposing solutions is outside its scope, maybe  prioritizing the pitfalls would help readers feel less overwhelmed or more easily identify concrete first steps that they can take. At the moment it is a flat, fairly long list of potential issues. All of the mentioned issues are important but perhaps outlining a hierarchy or suggesting some concrete actions would result in the paper having a bigger impact in terms of improving the generalization of ML systems used in practice.

It was not clear to me why the experiments use only a small portion of the BRATS dataset.

Finally, in my opinion the main weakness is that the paper lacks a clear distinction between causal inference and prediction, and I find the discussion of confounders misleading -- see more details in "required changes".

---

> ### Comment · Reviewer_UGJ4 · 2024-10-27
>
> I forgot to mention in my "minor comments": in a few places the authors mention that a hyperparameter was selected from a certain range using the validation set such as the learning rate. When 25 experiments are performed on 25 different datasets, in theory 25 different hyperparameters could have been selected? Only one is mentioned so I don't think I understood very well the details of how the training, model selection, and validation were organized in the experiments
>
> Also since the authors mention "double descent": it seems in some cases at least the phenomenon might be due to how the model complexity is defined and increased so this reference might be worth mentioning as well: [Curth & al 2023 "A U-turn on Double Descent: Rethinking Parameter Counting in Statistical Learning"]

---

> ### Author Response · Authors · 2024-11-09
>
> We thank you for the valuable feedback. Below, we address each of the concerns raised by the reviewer and have made the corresponding revisions to the manuscript.
>
> ## Requested Changes:
> **Comment1: Confounders and P3**
>
> We agree with the reviewer that the previous terminology implied a confusion between causal confounders and non-generalizable correlations in predictive modeling, and we thank you for pointing this out. We have revised the text to clarify that our focus is on spurious correlations in ML for which we give a clear definition and reference now. We also distinguish between different possible spurious correlations. We have now revised the entire section and the title is changed to P3-Spurious Correlations. The main changes are highlighted in the uploaded manuscript.
>
> **Comment2: Confounders and Experiment 1**
>
> We agree with the reviewer that, in this experiment, the noise is not a confounder but rather a spurious correlation that the model uses as a shortcut. We have revised this throughout Experiment 2, renaming it to "Spurious Correlations" and consistently referring to noise as a spurious correlation or shortcut rather than a confounder.
>
> Additionally, we acknowledge that this experiment includes sample selection bias. As noted earlier in the P3-Spurious Correlations section, spurious correlations can lead to generalization issues when there is a dataset shift. Therefore, we deliberately introduced this sample selection bias to create a scenario that could occur in real-world situations. The aim was to illustrate how such spurious correlations can lead to poor generalization or even the mistaken belief that the model is learning meaningful input-output relationships. We have now also added a note in the experiment to clarify that this sampling bias was introduced to create conditions where the model might rely on spurious correlations.
>
> **Comment3: Experimental design and ML in P3**
>
> Thank you, we have now added a discussion to the end of Section P1, where we now connect it to large-scale machine learning and the benefits that can be obtained in modern machine learning when attempting design of experiment through active learning or online adaptive experimental design.
>
> **Comment4: Discussion of the different levels of grouping across which a model is expected to generalize**
>
> We have added a discussion on the different levels of grouping and their implications for model generalization in the P2—Dataset Shift section. It is added to the second part of paragraph 7 of this section and starts with: "Therefore, it is important to determine the levels across which the model is expected to generalize."
>
> ## Minor comments:
>
> **Comment1: Class imbalance is not a dataset shift**
>
> We agree that class imbalance does not necessarily imply dataset shift if the prior probability of each class remains the same between training and target populations. To address this, we have removed the example of imbalanced labels from the "P2-Dataset Shift" section and moved it to the "P9-Misused Metrics" section, where we discuss how imbalanced labels can impact metric reliability, especially with metrics like accuracy.
>
> **Comment2: Why experiment on 3 pitfalls**
>
> Since it was not possible (or feasible) to experiment on all 14 pitfalls, we put focus on three common ones. We chose these specific pitfalls because we thought they were important, could be demonstrated well with a public dataset, and because we had concrete ideas for how to illustrate these points in an experiment. For other pitfalls, we did not have clear experiments ready. These selected experiments also allowed us to address related concepts, such as metrics and model interpretability.
>
> **Comment3: Why small portion of BraTS data**
>
> We chose to use a smaller subset of the BraTS dataset for two main reasons. First, for our site-based experiments, we needed enough patients from each hospital, but most hospitals in the dataset do not have enough patients for our study. Second, we wanted smaller experiments to make them faster and more feasible, especially in the case of experiment~1 where we did 25 repetitions of the experiment.
>
> **Comment4: MRI scans not slices**
>
> Thank you for catching this. Yes, they are brain scans and we have clarified this description of the data now.
>
> **Comment5: overlapping data and t-test**
>
> We have reviewed the dataset construction process and observed a degree of overlap (estimated at approximately 10\%) across the 25 datasets due to different random seeds. We agree that this overlap may introduce dependencies, potentially impacting the strict independence assumption of the paired t-test. To address this, we have updated the manuscript to acknowledge the overlap in "Results and discussion" section of Experiment~1. We have added a note advising cautious interpretation of the statistical results.

---

> ### Author Response · Authors · 2024-11-09
>
> **Comment6: Experiment 3 and leakage**
>
> We agree with the reviewer that comparing a case where normalization is done across the training set vs. the entire dataset would be interesting and could provide additional insights. However, implementing this would require not only changing the leakage-free setup (to normalize using training set statistics instead of each slice) but also adjusting the single-leakage setup for consistent preprocessing to be able to compare these cases. This would take additional time, and was not feasible within the two-week revision period.
>
> Also, the most common way to normalise MRI images is to use statistics from each slice or from a patient's MRI volume. Normalising based on the entire dataset is very unusual, and therefore not a realistic experiment. For that reason, we also used this normalizing approach in Experiments 1 and 2. We applied the same approach in the leakage-free setup in Experiment 3 to stay consistent and avoid introducing scenarios with different, potentially confusing details.
>
> Another option would be to add an entirely separate experiment, where "leakage-free" case uses training set statistics for normalization, and "leakage" case normalizes based on the whole dataset. However, we felt this might introduce too many scenarios and details, making the text complex and potentially confusing for readers. Also, this would likely require further input from the other reviewers as well, since it would mean to expand the manuscript.
>
> **Comment7: Experiment 3, prioritizing pitfalls, suggesting actionable steps**
>
> We agree with the reviewer about the suggested experiment. However, for the same reasons mentioned in the response of comment 7 conducting a new experiment would not be feasible now. We also agree that as the reviewer mentions in the "Weaknesses" section introducing actionable steps and prioritizing pitfalls would be valuable for practitioners. However, due to the complex and overlapping nature of these pitfalls, such prioritization requires extensive investigation, and it would also be difficult not to make it be subjective. To address this in part, we have relied on our review findings to suggest some actionable recommendations based on how common they were. We have added a new heading in Section 4, titled "Practical insights from review findings", presented as three paragraphs at the end of Section 4, just before the conclusion, to give some practical guidance based on commonly overlooked practices observed in our review.
>
> **Comment8: Some of the pitfalls and review questions do not relate directly to generalization...**
>
> We agree that code sharing alone does not immediately determine a model's capacity to generalize. However, reproducibility practices enhance transparency and give the ML community the opportunity to test a model's generalization under diverse conditions beyond the initial study's scope. A concrete example is when a new developed model is compared to existing ones in the literature. If the ones from the literature are not available for fine-tuning and/or re-training in the present conditions, the comparisons may be wrong and may affect our own modeling decisions, eventually also affecting our model's ability to generalize. Therefore, code sharing, although not directly, supports the broader goal of validating models' real-world applicability. We have added a short discussion to P14-Code Reproducibility Issues section to explain this point.
>
> **Comment9: 150 successful machine learning models: 6 lessons learned at booking.com**
>
> We find the paper related, and their experience with proxy over-optimization was especially interesting. We added a discussion on how focusing on short-term metrics like click-through rate in recommendation systems can lead to the "paradox of choice", where users are presented with similar options that increase clicks but ultimately harm satisfaction and conversion. This has been added into our discussion on short-term concerns in P9-Misused metrics section.
>
>
> **Comment10: hyperparameters in Experiment 1**
>
> Initially, our experiment was designed with a single dataset, and we did hyperparameter tuning on this dataset using its validation set. Later, we decided to extend our analysis to include 25 new datasets to have a broader understanding of generalization gaps. To keep consistency and to make the experiments feasible, we used the hyperparameters selected from the initial dataset across these 25 datasets, rather than re-tuning for each again.
>
> **Comment11: double descent**
>
> Thank you for the recommendation. We have now included a discussion in P8-Over-/Under-fitting section that explains how double descent may result from viewing model complexity along separate axes.

---

> > ### Comment · Reviewer_UGJ4 · 2024-12-03
> >
> > I would like to thank the authors for answering my questions and updating the manuscript and in particular section P3. I think that all my comments have been addressed. I only noted a typo on page 9: "when there is no dataset shifts".

---

> > > ### Author Response · Authors · 2024-12-05
> > >
> > > We appreciate your review of our paper, and especially your comment about section P3. Thank you for noting the typo. We have now fixed it in the new uploaded version.

---

### Decision · Action_Editor_gY2X · 2024-12-09

**Recommendation:** Accept with minor revision

**Comment:**

The contributions is very clear, and a nice addition to the literature.
60 publications are reviewed, leading to 14 pitfalls across 4 stages of the development of a ML pipeline. For 3 of these (dataset shift, confounders and data leakage), the authors show on the Brain Tumor Segmentation dataset how they can lead to wrong conclusions.
The quality of reporting is also evaluated, on 126 publications to ICCV and MICCAI.

**Audience:**

The paper serves as a useful tutorial/caveat list for ML practitioners. It provides many references to learn more about any of the 14 identified pitfalls. The paper may even be considered as a checklist to abide by when conducting a ML experiment (i.e., were the baseline hyperparameters tuned?).
Although it does not offer concrete solution, this work is nevertheless a very useful resource.

**Claims And Evidence:**

The paper provides a useful summary of the frequent pitfalls that may occur in a machine learning pipeline, and prevent models from generalizing well.
Several issues are highlighted, with practical examples from published papers serving as convincing examples.
The paper abundantly uses relevant references in its literature review.